# Agent JIT Compilation for Latency-Optimizing Web Agent Planning and Scheduling

**Caleb Winston** [1]   **Ron Yifeng Wang** [1]   **Azalia Mirhoseini** [1]   **Christos Kozyrakis** [1]

## Abstract

Computer-use agents (CUA) automate tasks specified with natural language such as "order the cheapest item from Taco Bell" by generating sequences of calls to tools such as click, type, and scroll on a browser. Current implementations follow a sequential fetch-screenshot-execute loop where each iteration requires an LLM call, resulting in high latency and frequent errors from incorrect tool use. We present agent just-in-time (JIT) compilation, a system that compiles task descriptions directly into executable code that is free to include LLM calls, tool calls, and parallelization. Our approach comprises three components: (1) JIT-Planner, which generates multiple code plans, validates each against tool specifications, and selects the minimum-cost candidate; (2) JIT-Scheduler, which explores parallelization strategies via Monte Carlo cost estimation from learned latency distributions; and (3) an invariant-enforcing tool protocol specifying precondition and postcondition requirements to reduce the rate of plans with incorrect tool use. Across five applications, JIT-Planner achieves $10.4\times$ speedup and 28% higher accuracy over Browser-Use, while JIT-Scheduler achieves $2.4\times$ speedup and 9% higher accuracy over OpenAI CUA.

## 1. Introduction

General automation of web-based tasks is an economically impactful AI application (Patwardhan et al., 2025). Frontier LLMs are capable of automating web-based tasks by predicting the next action given the sequence of prior web browser state and actions, where the state may be a screenshot image or text from the document object model (DOM) or accessibility tree. However, benchmarks such as We-

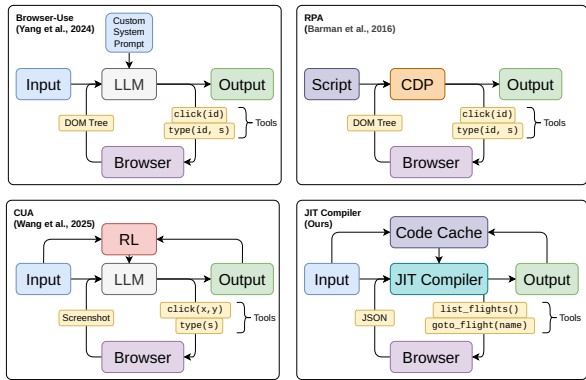

*Figure 1.* **Competing Approaches to Computer-Use Agents.** Automation of web-based tasks has relied on static scripts (RPA; Barman et al., 2016) and static tool sets (CUA; Wang et al., 2025). Our work introduces dynamic cost-optimizing planning and scheduling with cached, reusable tools.

bArena (Zhou et al., 2023), WebVoyager (He et al., 2024), and REAL (Garg et al., 2025) have shown that naively applying frontier LLMs yields poor latency and accuracy. Recent attempts to improve these include more aggressive filtering of the DOM or accessibility tree to feed the LM (Schiepanski & Piël, 2025) and reinforcement-learning to train smaller models for generating actions given web browser state (Wang et al., 2025; Wei et al., 2025; Bhathal & Gupta, 2025). However, these attempts continue to suffer from significant limitations across both latency and accuracy aspects:

**Static tool set.** The model selects from a static set of tools (e.g., click coordinates, type text) that is the minimum spanning set of actions to traverse all possible environment (web browser) states. While this generality lets the tool set reach any state, it comes at the cost of longer execution times and higher error rates.

**Sequential execution.** Prior work fails to explore the complete space of execution schedules, including task parallelism and request hedging (speculative parallel execution, returning the first valid result). For example, given a dynamic vCPU budget (e.g., 4), a task to "tell me about the 3 least expensive items from Taco Bell" admits a finite but nontrivial space of possible execution schedules: serially, in

[1]Stanford University, Stanford, CA, USA. Correspondence to: Caleb Winston <calebwin@stanford.edu>.

*Proceedings of the 43rd International Conference on Machine Learning*, Seoul, South Korea. PMLR 306, 2026. Copyright 2026 by the author(s).

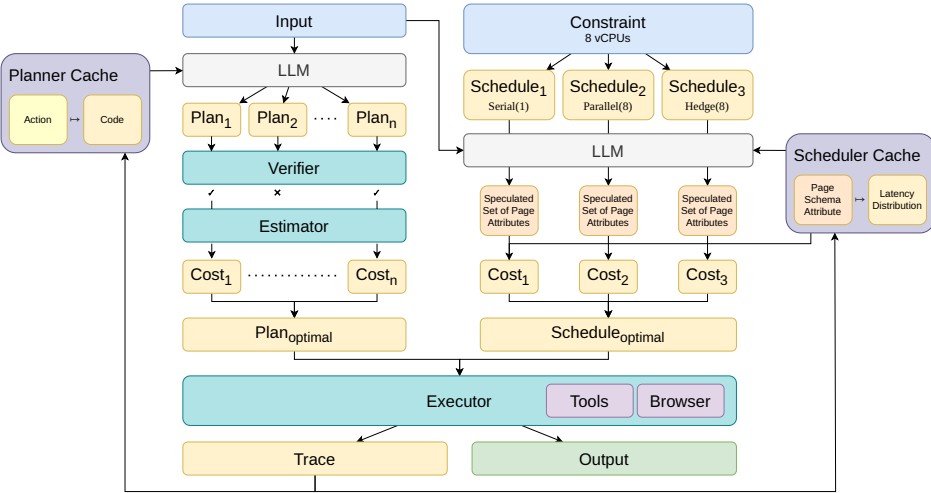

*Figure 2.* **Agent JIT Architecture.** Optimizing scheduling and planning for computer-use agents with caching of code and latency distributions.

parallel for each item, or with hedging.

**Non-determinism post-plan.** In the standard agent loop, once a plan is generated, execution may be unnecessarily non-deterministic post-planning. For example, a plan may specify that the output of a tool to list items is piped into a tool to order the least expensive item. A nondeterministic LM call is not strictly required between these two tool calls, but in a standard agent loop, the LM is called at each step.

We address these limitations via *agent just-in-time (JIT) compilation*. Like a JIT compiler, which translates a higher-level program into optimized lower-level code at run time, our system translates a natural-language instruction into a lower-level code plan at plan synthesis time. The plan orchestrates tool calls and inference, and among many valid translations we select the one with the lowest estimated cost. This code is built from cached, reusable tools (e.g., list_restaurants, add_to_cart) rather than primitive actions (e.g., click, type), so the LM need not be called at every step. Our contributions are the components of an agent JIT compiler for computer-use:

- **Cost-optimizing planner**: Plans are code, enabling parallel candidate generation as well as static checking and cost estimation over a control-flow graph (CFG) (Section 3.2, Algorithm 1).
- **Cost-aware scheduler**: Parallelization strategy selection via Monte Carlo cost estimation from prior learned latency distributions (Section 3.3, Algorithm 2).
- **Invariant-enforcing tool protocol**: Tools specify precondition and postcondition state invariants (Section 3.1), enabling compositional verification at compilation time.

We present our results in Section 5, demonstrating that JIT-Planner achieves $10.4\times$ speedup and $+28\%$ accuracy improvement over Browser-Use, while JIT-Scheduler achieves $2.4\times$ speedup and $+9\%$ accuracy improvement over OpenAI CUA.

## 2. Related Work

**Code actions.** Nam et al. (2024) first proposed unifying the action space of an LLM agent with a single tool to execute code. Song et al. (2025) has applied this technique to web automation, and Dong et al. (2025) surveys general application. However, prior work has not considered that for a given agent, there may exist many possible code plans with varying latency. In our work, we find that the difference between the best-latency and worst-latency code plan candidates is $5.3\times$ (Section 5). Hence, we propose an optimizing JIT compiler that translates a task into the code candidate with the optimal estimated cost.

**Web agents.** Web-browsing agents generate sequences of actions that actuate a web-browsing environment. Previous work has heavily studied two key approaches to implementing web agents: (1) prompt-engineering of frontier LLMs to output actions given browser state (Yang et al., 2024), and (2) fine-tuning smaller models to output actions given browser screenshots (Wang et al., 2025; Wei et al., 2025). Our work instead focuses on modifying the set of actions to include cached, reusable code actions and optimizing the planning and scheduling of these actions. Figure 1 positions our work relative to prior web automation approaches.

**Agent latency optimization.** Existing work optimizing the latency-accuracy trade-off of LLM agents has focused on routing (Jiang et al., 2025) or caching (Bian et al., 2025) at the model level. In contrast, we focus on optimizing the

latency-accuracy trade-off at the level of tool-use action sequences via cost-aware planning and scheduling.

**Tool protocols.** Tool protocols have been proposed to specify interface contracts for LLM agents. The Model Context Protocol (MCP) (Ray, 2025; Lumer et al., 2025) has emerged as a standard for type-checking of tool inputs and outputs. In our work, we extend this concept to invariant-enforcing tool protocols that specify precondition and postcondition state invariants, enabling compositional verification of tool sequences at compilation time.

## 3. Methods

In this section, we provide an end-to-end overview of cost-optimizing JIT compilation for computer-use agents. As shown in Figure 2, the architecture consists of three main components: (1) a cost-optimizing planner, (2) a cost-aware scheduler, and (3) an invariant-enforcing tool protocol. An offline process collects execution traces to update two caches: (a) a planner cache mapping actions to reusable tool code, and (b) a scheduler cache mapping schema elements to latency distributions (Section A).

### 3.1. Invariant-Enforcing Tool Protocol

A key challenge in generating plans for computer-use agents is ensuring correctness of tool sequences. We introduce an invariant-enforcing tool protocol where each tool specifies precondition and postcondition state invariants. The protocol requires each tool implement the following:

| Field | Description |
|---|---|
| pre | Expected state before execution. |
| pre_check | Optional runtime predicate. |
| post | Guaranteed state after execution. |
| post_check | Optional runtime predicate. |
| input_schema | Type constraints on parameters. |
| output_schema | Type constraints on return values. |
| execute | Implementation code. |

For example, a protocol-compliant tool to navigate to a restaurant list page may specify:

```
{
 "input_schema": {"rId": "string"},
 "output_schema": {},
 "pre": {"page": "*"},
 "post": {"page": "detail",
          "selectedRestaurant": "$rId"},
 "pre_check": "document.getElementById(
               'r-list') != null",
 "post_check": "document.getElementById(
                'r-detail') != null",
 "execute": "document.getElementById(
             'r-'+rId).click();"
}
```

Implementing this protocol enables tool use to be statically checked for (1) type safety according to the tool signa-

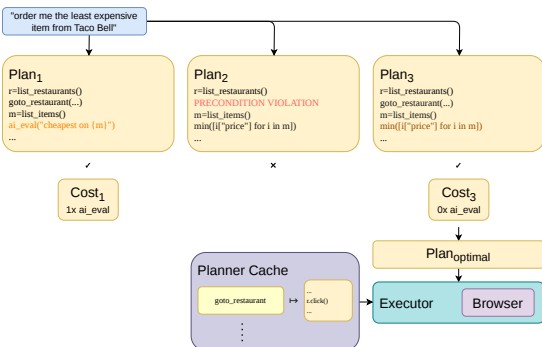

*Figure 3.* **Example with planner.** Three plans generated in parallel: Plan₁ has an unnecessary LM call, Plan₂ violates the precondition of `list_items`, Plan₃ is cost-optimal as it replaces an LM call with code.

ture and (2) state flow correctness according to preconditions and postconditions, where tool calls are composable if $post_i \subseteq pre_{i+1}$. The optional runtime predicates (`pre_check`, `post_check`) allow for runtime verification of preconditions and postconditions for debugging.

The motivation for state flow checking is our finding (Section 5) that 45-50% of errors in web automation arise from incorrect sequences of actions (e.g., clicking wrong element, typing into wrong field). By enforcing state flow correctness at compilation (plan synthesis) time, we eliminate a large class of errors.

### 3.2. Cost-Optimizing Planner

While an invariant-enforcing tool protocol ensures correctness of a tool sequence plan, there may exist many valid plans for a given task with varying latency and accuracy trade-offs. We propose a cost-optimizing planner that generates multiple candidates in parallel, validates them via static checking, estimates their costs via CFG traversal, and selects the minimum-cost plan (Algorithm 1).

Figure 3 shows parallel plan generation where Plan₃ is selected as optimal. Despite Plan₂ using fewer LLM calls than Plan₁ (0× vs 1× `ai_eval`), it violates state flow constraints. The planner's CFG traversal simultaneously validates state transitions and estimates cost (Algorithm 1, lines 7-22), rejecting invalid plans early.

**Offline tool synthesis.** The planner cache illustrated in Figure 11 is populated via LLM synthesis of tool code from execution traces.

**Parallel candidate generation.** (Algorithm 1, lines 5-24): $n$ workers independently sample code plans from model $M$ in parallel until $k$ valid candidates are collected (line 5). Each worker iteratively refines failed plans using validation errors as feedback (line 6). Early acceptance terminates

**Algorithm 1** Cost-Optimal Planning

1: **Input:** Task $\tau$, Manifests $\mathcal{M}$, Model $M$, Workers $n, k$
2: **Output:** Plan $\pi^*$ or $\varnothing$
3: **Constants:** Costs $C_{\text{tool}}$, $C_{\text{eval}}$, Decay $\gamma$
4: $C \leftarrow \emptyset$
5: **for** $i = 1$ to $n$ **parallel until** $|C| \geq k$ **do**
6:     **for** $iter = 1$ to $M_{\max}$ **do**
7:         $\pi \leftarrow M(\tau, \mathcal{M})$; $CFG \leftarrow \text{BuildCFG}(\pi)$
8:         $s \leftarrow \emptyset$, valid $\leftarrow$ true, cost $\leftarrow 0$
9:         **for each** block $B$ in $CFG$ at depth $d$ **do**
10:             **for each** call $c$ in $B$ **do**
11:                 **if** $c$ is Tool $t$ **then**
12:                     **if** $s \not\supseteq \mathcal{M}[t].\text{pre}$ **then**
13:                         valid $\leftarrow$ false; **break**
14:                     $s \leftarrow s \cup \mathcal{M}[t].\text{post}$
15:                     cost $+= C_{\text{tool}} \cdot \gamma^d$
16:                 **else if** $c$ is AI_EVAL **then**
17:                     cost $+= C_{\text{eval}} \cdot \gamma^d$
18:                 **end if**
19:             **end for**
20:             **if** !valid **then break**
21:         **end for**
22:         **if** valid **then** $C \leftarrow C \cup \{(\pi, \text{cost})\}$; **break**
23:     **end for**
24: **end for**
25: **return** $\text{argmin}_{(\pi,c)\in C} c$

when $k$ valid candidates are collected, reducing latency.

**CFG-based validation and cost estimation.** (Algorithm 1, lines 7-22): For each plan $\pi$, we build a CFG (line 7) and traverse it to simultaneously validate state flow and estimate cost (lines 9-21). For each tool call at depth $d$, we verify $state \supseteq pre$ (lines 12-13), update state with postconditions (line 14), and accumulate cost $C_{\text{tool}} \cdot \gamma^d$ (line 15). For ai_eval calls, we accumulate $C_{\text{eval}} \cdot \gamma^d$ (line 17). The nesting penalty $\gamma = 10$ penalizes nested iterations over expensive LLM calls, guiding selection toward efficient plans that minimize ai_eval usage in loops. The minimum-cost valid plan is returned (line 25).

### 3.3. Cost-Aware Scheduler

The scheduler evaluates three vCPU allocation strategies: serial (sequential execution), parallel (task parallelism across $n$ workers), and hedge (first-of-$k$ redundant execution). Using latency distributions learned offline (Figure 11) from prior element interactions, the scheduler predicts which DOM elements the plan will interact with, then estimates expected latency via Monte Carlo sampling (Algorithm 2).

**LLM-based element usage prediction.** (Algorithm 2, line 5): For each strategy $\sigma \in \{\text{SER}, \text{PAR}, \text{HED}\}$, model $M$ predicts element usage patterns $U_\sigma$ given task $\tau$, cached elements $\mathcal{E}$, and strategy $\sigma$. This yields element-to-count mappings for serial execution and per-worker element assignments for parallel strategies.

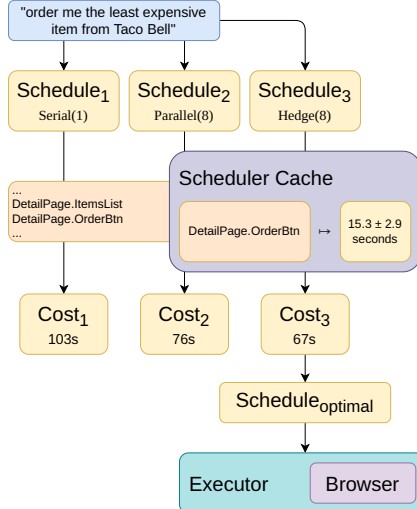

*Figure 4.* **Example with scheduler.** Three strategies are evaluated for each given task: serial execution, task-parallelism, and request hedging. Monte Carlo sampling from learned latency distributions for different elements in the web browser environment yields request hedging as optimal given the high latency variance of interaction with the order button.

**Algorithm 2** Cost-Aware Scheduling

1: **Input:** Task $\tau$, Cache $\mathcal{E}$, Dists $\mathcal{D}$, Model $M$, Workers $n$
2: **Output:** Strategy $\sigma^* \in \{\text{SER}, \text{PAR}, \text{HED}\}$
3: **Constants:** Trials $N_{\text{MC}}$, Overhead $\delta_p$ (Par), $\delta_h$ (Hedge)
4: **for each** strategy $\sigma \in \{\text{SER}, \text{PAR}, \text{HED}\}$ **do**
5:     $U_\sigma \leftarrow M.\text{PREDICTUSAGE}(\tau, \mathcal{E}, \sigma)$
6:     $L_\sigma \leftarrow \emptyset$
7:     **for** $trial = 1$ to $N_{\text{MC}}$ **do**
8:     **if** $\sigma = \text{SER}$ **then**
9:       $\ell \leftarrow S(U_\sigma)$
10:     **else if** $\sigma = \text{HED}$ **then**
11:       $\ell \leftarrow \min_{w=1}^{n} S(U_\sigma) + \delta_h$
12:     **else if** $\sigma = \text{PAR}$ **then**
13:       $\ell \leftarrow S(U_\sigma^{\text{seq}}) + \max_{w=1}^{n} S_w(U_{\sigma,w}) + \delta_p$
14:     **end if**
15:     $L_\sigma \leftarrow L_\sigma \cup \{\ell\}$
16:     **end for**
17:     $\bar{L}_\sigma \leftarrow \text{mean}(L_\sigma)$
18: **end for**
19: **return** $\text{argmin}_\sigma \bar{L}_\sigma$

**Monte Carlo cost estimation.** (Algorithm 2, lines 4–14): To estimate latencies, we sample from learned distributions using $S(U) = \sum_{e \in U} \sum_{i=1}^{\text{count}(e)} \text{sample}(\mathcal{D}_e)$, where $\text{count}(e)$ is the predicted number of interactions with element $e$. For each strategy, we run $N_{\text{MC}}$ trials (line 7): (1) serial cost as $S(U_\sigma)$ (line 9), (2) hedge cost as the minimum across $n$ redundant workers plus overhead $\delta_h$ (line 11), (3) parallel cost as the sequential portion plus the maximum bottleneck worker plus overhead $\delta_p$ (line 13). The strategy with minimum mean latency $\bar{L}_\sigma$ is returned (line 19). Figure 4 illustrates the three strategies.

| | | Browser-Use | | Browser-Use +cache | | JIT-Planner | | |
|---|---|---|---|---|---|---|---|---|
| | | Latency | Accuracy | Latency | Accuracy | Latency (s) | Range (s) | Accuracy |
| Model | GPT-4.1 | 150.1 | 61% | 105.2 | 88% | **15.4** (9.7x) | 93.7 | **90%** (+29%) |
| | Gemini-2.5-Flash | 100.3 | 59% | 69.3 | 81% | **7.2** (14.0x) | 22.4 | **94%** (+35%) |
| | Gemini-2.5-Pro | 115.9 | 77% | 65.8 | 86% | **12.6** (9.2x) | 33.8 | **97%** (+20%) |
| Task Cardinality | C-Low | 101.5 | 64% | 65.2 | 86% | **9.4** (10.8x) | 36.3 | **94%** (+30%) |
| | C-Medium | 119.0 | 75% | 91.2 | 85% | **13.7** (8.7x) | 26.1 | **94%** (+19%) |
| | C-High | 169.0 | 58% | 96.2 | 84% | **14.3** (11.8x) | 93.1 | **92%** (+33%) |
| Task Length | T-Short | 70.2 | 73% | 51.5 | 88% | **7.1** (10.0x) | 26.6 | **98%** (+24%) |
| | T-Medium | 134.9 | 59% | 86.8 | **93%** | **13.0** (10.4x) | 35.5 | 92% (+33%) |
| | T-Long | 195.3 | 62% | 121.2 | 70% | **18.4** (10.6x) | 91.6 | **89%** (+27%) |
| Application | Dashdish | 118.8 | 65% | 62.8 | **100%** | **14.5** (8.2x) | 35.3 | **100%** (+35%) |
| | GitLab | 144.7 | 49% | 69.7 | 67% | **9.9** (14.6x) | 20.7 | **87%** (+38%) |
| | Gomail | 133.8 | 63% | 94.6 | 80% | **10.4** (12.9x) | 24.9 | **96%** (+33%) |
| | Omnizon | 113.9 | 76% | 86.8 | 84% | **15.5** (7.4x) | 93.6 | **89%** (+13%) |
| | Reddit | 81.3 | 72% | 82.7 | 89% | **4.3** (18.9x) | 8.5 | **93%** (+21%) |

*Table 1.* **End-to-end results across methods.** Latency (seconds), Range (seconds), and Accuracy (%) aggregated by model, task complexity, task length, and application. *Range* is the latency gap between the worst-cost and best-cost candidate plans (i.e., the best-cost latency plus Range equals the worst-cost latency). Speedup and improvement are shown in parentheses for JIT-Planner.

## 4. Evaluation

### 4.1. Tasks

**Task Complexity.** We evaluate across tasks with varying complexity along two dimensions: cardinality and length. *Task cardinality* measures the number of elements processed, analogous to database query cardinality (C-Low: 1 element, C-Medium: 2–4 elements, C-High: 5+ elements). *Task length* measures the number of steps required under baseline Browser-Use (T-Short: 1–5 steps, T-Medium: 6–8 steps, T-Long: 9+ steps). These thresholds are determined by terciles (33rd and 67th percentiles) of each distribution. Tasks span both dimensions independently; high cardinality does not imply long execution.

**Applications.** Our evaluation spans five applications from two benchmarks, covering diverse web interaction patterns:

*REAL* (Garg et al., 2025): (1) **Dashdish** is a DoorDash-inspired food delivery platform testing restaurant browsing, filtering, pricing extraction, and order placement. (2) **Go-mail** is a Gmail-inspired email interface evaluating inbox navigation, message filtering, and email composition. (3) **Omnizon** is an Amazon-inspired e-commerce platform testing product search, comparison shopping, cart management, and checkout flows. For each REAL application, we select three tasks from the benchmark and manually curate six additional tasks, ensuring that across the nine tasks per application, three are optimal under each of serial, parallel, and hedge scheduling strategies.

*WebArena* (Zhou et al., 2024): (4) **GitLab** is a code repository platform testing project navigation, issue management, and settings modification. (5) **Reddit** (Postmill) is a social discussion platform testing post navigation, content filtering, and comment interaction. For each WebArena application, we randomly select five tasks from the benchmark. These tasks are evaluated with the planner only (without scheduling), as WebArena tasks are primarily sequential.

The REAL benchmark harness uses state-diff checking to validate task completion, comparing initial and final application states to verify objective achievement. WebArena's harness uses pre-defined evaluation functions. Together, these five applications span 37 tasks covering e-commerce, communication, collaboration, and social domains. Full task descriptions and oracle scheduling strategies are provided in Section D.

### 4.2. Methods and Baselines

We compare against several baselines spanning different architectural approaches.

**Computer-Use Agents (CUA).** Computer-use agents from foundation model providers, which select actions from a fixed tool set and execute serially. We evaluate Anthropic CUA (Claude Sonnet 4 with computer-use tools) and OpenAI CUA (GPT-4o fine-tuned via reinforcement learning for computer interaction).

**Browser-Use.** Standard agentic approach with minimal tooling (Yang et al., 2024): the LLM generates the next action at each step, executes via browser automation, observes the result, and repeats. This represents the baseline agent loop without code generation or planning. We evaluate Browser-Use, Browser-Use +cache, and our JIT methods with the same models (GPT-4.1, Gemini-2.5-Flash, Gemini-2.5-Pro).

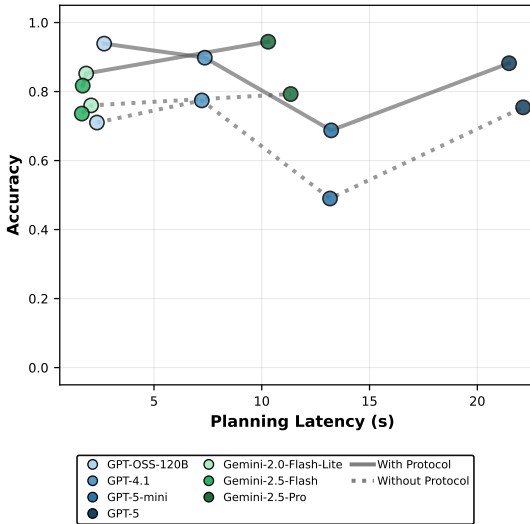

*Figure 5.* **Impact of protocol on latency-accuracy frontier in plan generation.** Usage of protocol-adherent manifests for each tool shifts the latency-accuracy Pareto frontier across different LMs sampled for tool-use plans.

**Browser-Use +cache.** Browser-Use augmented with synthesized tools generated via our caching mechanism (Figure 11). Successful action traces are converted to reusable code tools, reducing redundant exploration. This approach is similar to tool discovery methods explored in WALT (Prabhu et al., 2025) but lacking the invariant-enforcing protocol and cost-optimizing planning and scheduling aspects.

**CUA +cache.** A frontier computer-use agent (GPT-5.4) given access to the same synthesized tool set as Browser-Use +cache and JIT-Planner. GPT-5.4 is a general-purpose model with native computer-use capabilities that achieves state-of-the-art results on computer-use benchmarks (OpenAI, 2026). It can call the cached tools but retains a standard per-step agent loop rather than cost-optimizing planning.

**JIT-Planner.** Cost-optimizing planning, as described in the agent JIT compiler design in Section 3.2.

**Fixed-Scheduler.** Static scheduling strategies that always use the same vCPU allocation approach: Serial (sequential execution), Parallel (task parallelism), or Hedge (request hedging with early termination).

**JIT-Scheduler.** Cost-aware scheduling, as described in Section 3.3.

**Oracle-Scheduler.** An oracle scheduler with perfect knowledge of true latencies. Represents an upper bound on scheduler performance for optimal latency.

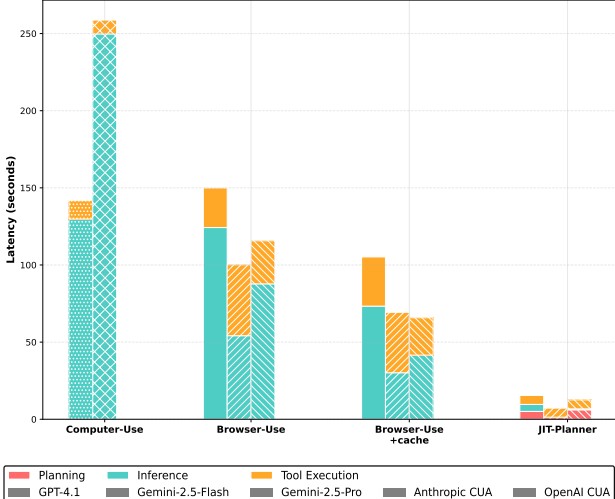

*Figure 6.* **End-to-end latency breakdown.** Comparison of latency across baselines and ablations with breakdown across Planning (LM sampling for plan generation and checking), Inference (LM calls during execution), and Tool Execution (web browser actuation including CDP and DOM operations).

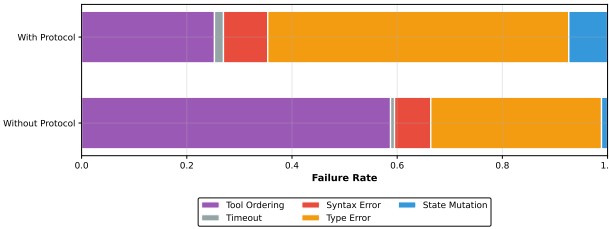

*Figure 7.* **Failure type shift with protocol enforcement.** Tool ordering dominates failures in generating tool-use plans. Protocol reduces total failure rate from 80% to 43% while shifting failure distribution.

## 5. Results

### 5.1. JIT-Planner

We evaluate cost-optimizing planning with respect to (i) planning accuracy and overhead, and (ii) end-to-end latency and accuracy.

**RQ1: Does an invariant-enforcing protocol improve planning accuracy?** We find that the use of the invariant-enforcing protocol introduced in Section 3.1 positively shifts the latency–accuracy Pareto frontier across all models we evaluate for plan generation (Figure 5). The improvement in valid-plan rate is statistically significant for the three models in our end-to-end evaluation (Section C): GPT-4.1 improves from 78% to 91%, Gemini-2.5-Pro from 79% to 96%, and Gemini-2.5-Flash from 74% to 85%. We further analyze failure modes in plan generation. Across models, tool-ordering violations (failures to satisfy a tool's precondition or postcondition) are the dominant failure mode without protocol,

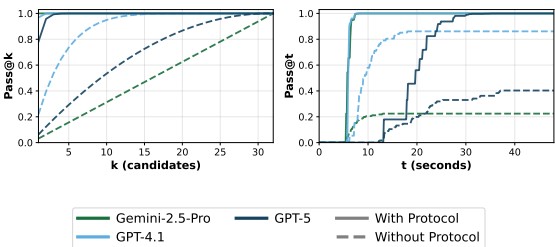

*Figure 8.* **Planning efficiency via Pass@k and Pass@t metrics.** On a long-horizon GitLab task (19 steps), protocol enforcement yields higher valid-plan rates with fewer candidates with fewer candidates (Pass@$k$, left) and within lower latency budgets (Pass@$t$, right) across frontier LMs (Gemini-2.5-Pro, GPT-4.1, GPT-5).

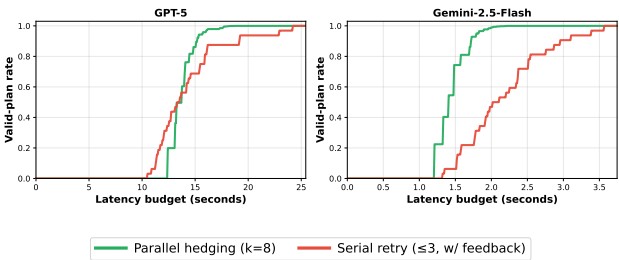

*Figure 9.* **Parallel hedging versus serial retry for plan generation.** Across latency budgets, protocol enforcement combined with parallel hedging (8 workers, accept first valid) outperforms serial retry (max 3 iterations) for both GPT-5 (left) and Gemini-2.5-Flash (right).

accounting for 59% of all failures versus 25% with protocol (Figure 7). This demonstrates that an invariant-enforcing protocol substantially improves the accuracy of tool-use planning.

**RQ2: Does it reduce planning overhead?** We introduce two metrics to quantify planning efficiency: Pass@$k$ and Pass@$t$. Pass@$k$ measures the probability of obtaining at least one valid plan in $k$ sampled candidates. Given $n$ total runs with $c$ successes,

$$\text{Pass@}k = 1 - \binom{n-c}{k}\Big/\binom{n}{k}$$

Pass@$t$ extends this to the temporal domain, capturing the probability of generating a valid plan within a latency budget $t$ when sampling candidates in parallel across $n_{\text{parallel}}$ workers:

$$\text{Pass@}t = 1 - (1 - F(t) \cdot p)^{n_{\text{parallel}}}$$

where $F(t)$ is the latency CDF and $p = c/n$ is the success rate.

We find that our invariant-enforcing protocol improves both metrics. On a long-horizon GitLab task (19 steps), the protocol increases Gemini-2.5-Pro's Pass@3 from 9% to 100%; under parallel hedging over plan generation with 8 workers, Pass@$t$ reaches 100% within approximately 8 seconds,

whereas without the protocol it plateaus at 22% (Figure 8). The same pattern holds for the other models: Pass@3 improves from 54% to 100% for GPT-4.1 and from 18% to 99% for GPT-5, with Pass@$t$ plateaus without the protocol of 86% and 40% respectively, both reaching 100% with it. This benefit is consistent across long-horizon tasks: aggregated over all nine T-Long tasks ($\geq 9$ steps), the protocol raises Gemini-2.5-Pro's valid-plan rate (Pass@1) from 58% to 93%.

We further compare parallel hedging (8 workers, return first valid) against serial retry (up to 3 sequential, feedback-refined attempts) on the same task (Figure 9). With the protocol, both strategies eventually reach a 100% valid-plan rate, but parallel hedging reaches the 95% threshold at a substantially lower latency budget: 16s versus 23s for GPT-5 and 1.8s versus 3.4s for Gemini-2.5-Flash. Parallel hedging therefore reaches high valid-plan rates at substantially lower latency budgets than serial retry.

**RQ3: Does cost estimation rank plans by latency and improve end-to-end performance?** We evaluate whether CFG-based cost estimation reliably ranks plans by actual latency, and whether cost-aware plan selection reduces end-to-end latency. Comparing minimum-cost versus maximum-cost plans reveals a substantial latency gap: JIT-Planner achieves 11.7s mean latency with best-cost selection versus 61.7s with worst-cost selection, a $5.3\times$ difference (Table 1). Because a few high-variance tasks inflate the mean, we also report the per-task paired ratio, which is $1.8\times$ (Section C). The gap between best- and worst-cost plans is mainly due to LLM inference and loop nesting, as shown in Figure 6, validating our cost model's penalties on LLM calls and nested loops. This confirms that cost estimation ranks plans effectively, even though it targets relative ordering rather than absolute latency (Section H).

Averaged across models, JIT-Planner (11.7s) achieves a $10.4\times$ speedup over Browser-Use (122.1s) and $6.8\times$ over Browser-Use +cache (80.1s) largely by eliminating unnecessary LLM calls, which account for 73% of Browser-Use's latency. Browser-Use +cache yields a $1.5\times$ improvement over Browser-Use but still runs an agentic loop with an LLM call at each step, whereas JIT-Planner compiles tasks to code.

**RQ4: Does task complexity affect planner performance?** We analyze performance across task complexity dimensions. Across task cardinality, JIT-Planner achieves $10.8\times$ speedup for C-Low, $8.7\times$ for C-Medium, and $11.8\times$ for C-High tasks (Table 1). We hypothesize that this non-monotonic pattern arises because medium-cardinality tasks have less variation in latency across plan candidates, leaving little room for cost-based selection, whereas high-cardinality tasks involve more tool calls and thus more opportunities to find a lower-cost plan. Across task length, JIT-Planner maintains consis-

tent speedup: 10.0× for T-Short, 10.4× for T-Medium, and 10.6× for T-Long. This indicates that the speedup derives primarily from eliminating per-step LLM inference rather than from task-specific optimization. Accuracy remains consistently high (89–98%) across both complexity dimensions, indicating that invariant enforcement generalizes across task characteristics.

**RQ5: How does performance vary across applications?**
We observe substantial variance across application domains, with speedups ranging from 7.4× (Omnizon) to 18.9× (Reddit), a 2.6× difference (Section B). Reddit achieves the highest speedup because its tasks compile to largely deterministic tool sequences, allowing the planner to eliminate per-step LLM inference almost entirely. Omnizon, by contrast, involves tasks requiring runtime semantic judgment (e.g., selecting a product that matches a request), which forces the plan to retain `ai_eval` calls and limits the achievable speedup. Accuracy improvements also vary: GitLab, with its diverse page types and many possible actions per page, shows the largest gain (+38%). Omnizon, with a relatively simple site structure, shows the smallest accuracy improvement (+13%) despite high baseline accuracy (76%). Together, these results suggest that JIT compilation benefits applications most when tasks can be compiled to deterministic tool sequences with minimal retained LLM inference, where eliminating per-step inference both reduces latency and removes a source of execution-time nondeterminism.

**Isolating planning from tool access.** To confirm that JIT-Planner's improvements are not solely attributable to the synthesized tools, we evaluate CUA +cache (Section 4.2), which gives GPT-5.4 access to the same tools as Browser-Use +cache and JIT-Planner, on the three REAL applications (27 tasks). Even with identical tools, CUA +cache remains 1.5–2.4× slower than JIT-Planner at comparable accuracy across all cardinality and length tiers (Section B.1), despite GPT-5.4 being a more capable, computer-use-native model than the LLMs JIT-Planner uses. This isolates the contribution of cost-optimizing planning from tool access.

## 5.2. JIT-Scheduler

We evaluate cost-aware scheduling under a budget of 4 vCPUs, comparing three static strategies — Serial (sequential execution), Parallel (task parallelism across workers), and Hedge (running redundant workers and taking the first to finish) — against adaptive selection.

**RQ1: How does execution strategy impact latency?** Latency varies substantially across static scheduling strategies, validating the need for adaptive selection. For GPT-4.1, Serial achieves 157.3s mean latency, Parallel 166.2s, and Hedge 130.3s, spanning a 1.3× range between best and worst static strategies (Figure 10). For Gemini-2.5-Pro, the corresponding values are 129.6s, 148.5s, and 98.4s, a 1.5×

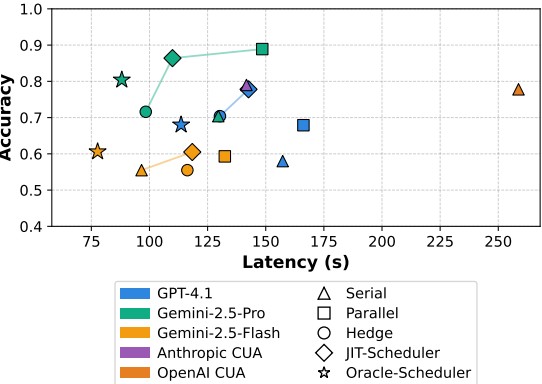

*Figure 10.* **Accuracy-latency trade-offs across scheduling strategies and models.** Monte Carlo cost estimation in JIT-Scheduler enables adaptive strategy selection that achieves favorable accuracy-latency trade-offs, outperforming static strategies and baseline CUA models. Lines trace the Pareto frontier over the schedulable strategies (Serial, Parallel, Hedge, JIT-Scheduler) per model. Oracle-Scheduler is shown as an upper-bound reference.

range. No single strategy is universally best, because the optimal choice depends on task structure in ways that are hard to predict. We explain this using specific examples from our evaluation. Parallel benefits tasks with independent subtasks, such as "get the first three reviews from each vendor," but its overhead can outweigh the gains; when subtasks are simple, Serial finishes faster, as for "how much is a matcha croissant and a regular iced matcha latte?," which requires only lookups on a single page. Hedge helps when individual workers are prone to getting stuck on difficult UI elements, so first-of-$n$ execution is more likely to find a fast path; this favors tasks such as "forward the first two emails and delete the originals" or those with uncertain success, such as "buy any taco." Finally, Serial is best for short, linear tasks, such as "how many restaurants offer delivery?" These trade-offs are difficult to predict from task structure alone, motivating cost-aware strategy selection.

**RQ2: Does JIT-Scheduler achieve favorable accuracy-latency trade-offs?** JIT-Scheduler uses Monte Carlo cost estimation over learned latency distributions to select a strategy per task. This achieves a favorable balance of latency and accuracy: For GPT-4.1, JIT-Scheduler achieves 142.6s with 77.8% accuracy, compared to Serial (157.3s, 58.0%), Hedge (130.3s, 70.4%), and Parallel (166.2s, 67.9%). For Gemini-2.5-Pro, JIT-Scheduler achieves 109.9s with 86.4%, compared to Serial (129.6s, 70.4%), Hedge (98.4s, 71.6%), and Parallel (148.5s, 88.9%). While individual fixed strategies may achieve lower latency (e.g., Hedge) or higher accuracy on specific models (e.g., Parallel for Gemini-2.5-Pro), JIT-Scheduler consistently achieves strong performance across both dimensions without requiring manual strategy selection. This accuracy advantage arises because the scheduler selects Parallel when subtasks are predicted

to be fast, thereby decomposing work into simpler pieces that are individually reliable. In contrast, Hedge executes the full task across workers, requiring each to complete more complex work. Compared to Oracle-Scheduler, which greedily minimizes latency, JIT-Scheduler achieves 6–10% higher accuracy at a 24.7–25.5% latency cost, demonstrating that cost-aware strategy selection effectively balances the latency-accuracy trade-off (Figure 10).

**RQ3: Does cost-aware scheduling improve upon baseline CUA?** Prior computer-use agents execute serially, without parallelism or hedging. On the three REAL applications, OpenAI CUA achieves 258.7s latency with 77.8% accuracy, while Anthropic CUA achieves 141.7s with 79.0% accuracy (per-application results in Section B). JIT-Scheduler achieves 109.9s with 86.4% accuracy for Gemini-2.5-Pro, representing a 2.4× speedup and +9% accuracy improvement over OpenAI CUA. Notably, while Anthropic CUA achieves similar accuracy (79.0%), it does so at 1.3× the latency of JIT-Scheduler. The performance gap demonstrates that cost-aware scheduling substantially improves both latency and accuracy over approaches that rely on serial execution, even for dedicated computer-use agents.

## 6. Limitations

**Offline setup cost.** The system requires a one-time offline setup per application: tool synthesis takes 25–90 min and trace collection for the scheduler cache takes 25–45 min, although both can be reduced to 20–30 min per application with parallel workers. These costs are amortized over all subsequent tasks, but make our approach best suited to applications an agent will handle repeatedly, rather than one-off interactions with novel applications.

**Cache staleness under UI changes.** Synthesized tools encode assumptions about an application's DOM structure, so UI changes (for example, due to redesigns or A/B tests) can break cached tools. The invariant-enforcing protocol detects this at runtime: a failed pre/postcondition marks the tool invalid, and the system falls back to re-planning without it. Other proactive strategies exist, such as TTL-based expiration or periodic re-validation.

**Stochastic environments.** Preconditions and postconditions check observable outcomes (e.g., "did the page navigate to the orders page?") rather than exact UI state, making them robust to incidental variation. However, highly stochastic environments (e.g., bot detectors, CAPTCHAs, rate limiting) can cause spurious postcondition failures, in which case the system falls back to generic tools and loses the latency benefit of cached tools. Extending the protocol to account for expected stochasticity is future work.

**Evaluation scope.** Our evaluation covers 37 tasks across 5 web applications from 2 benchmarks, varying cardinality and length with 3 trials per configuration. Although we have not evaluated desktop or mobile environments, the approach is not fundamentally limited to the web. The key requirement is programmatic access to application state for precondition and postcondition evaluation, which is often available via accessibility APIs on desktop platforms. Extending to additional environments is a meaningful direction for future work.

## 7. Conclusion

In this paper, we presented agent JIT compilation, a system for executing computer-use agents by dynamically translating agent tasks into code. We introduced an invariant-enforcing tool protocol and cost-optimizing planning and scheduling. Our results showed: First, invariant-enforcement yields tool-use plans that are generated more accurately and with lower overhead. Second, cost-optimizing planning can improve the latency-accuracy Pareto frontier for computer-use agents (10.4× speedup and 28% accuracy improvement over Browser-Use), especially for longer or more complex tasks. Cost estimation accurately selects lower latency plans where poor selection could otherwise degrade latency by as much as 5.3× on average. Third, cost-aware scheduling yields similar benefits, outperforming OpenAI's CUA model with a 2.4× speedup and 9% accuracy improvement. In conclusion, agent JIT compilation presents a promising direction for improving the efficiency and reliability of computer-use agents.

## Impact Statement

This work makes computer-use agents that automate web tasks faster, cheaper, and more reliable. Beneficial uses include reducing repetitive digital labor and improving accessibility for users who struggle with conventional interfaces. Compilation of agent tasks to code also reduces the LLM inference per task, lowering the compute and energy cost relative to per-step agent loops. The same gains in throughput and reliability could also lower the cost of abusive automation such as spam, fraud, or scraping behind authentication. These effects are uncertain and depend heavily on how, and by whom, such systems are deployed.

A central direction of this work is relevant to the safe deployment of such systems. LLMs increasingly generate autonomous, dynamically composed workflows that execute without verification, making their behavior hard to predict or audit. Our approach instead annotates tools with declared invariants and statically checks each generated plan against them before execution, turning an opaque per-step decision process into an inspectable, verifiable artifact. We view invariant annotation and pre-execution verification of generated plans, including runtime enforcement of precon-

ditions before irreversible actions, as a tractable direction for governing increasingly autonomous agents.

Our experiments used sandboxed benchmark environments (REAL and WebArena) and self-hosted deployments, with no real user data or live third-party services.

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

## A. Offline Caching Architecture

Figure 11 illustrates the offline process that populates the planner and scheduler caches from execution traces. The pipeline consists of three stages:

1. **Extract.** Each trace step is processed by an LLM to extract a page schema and identify the high-level action performed. The page schema defines the actionable elements on each page (Section I, Prompt 3).
2. **Label.** Each trace step is mapped to a page schema element, producing (element, latency) observations. These observations are aggregated and fit into per-element latency distributions, populating the scheduler cache (element $\mapsto$ distribution) (Section I, Prompt 4).
3. **Codegen.** A synthesis agent uses the extracted actions and page context to generate protocol-compliant tool code, populating the planner cache (action $\mapsto$ code) (Section I, Prompt 5).

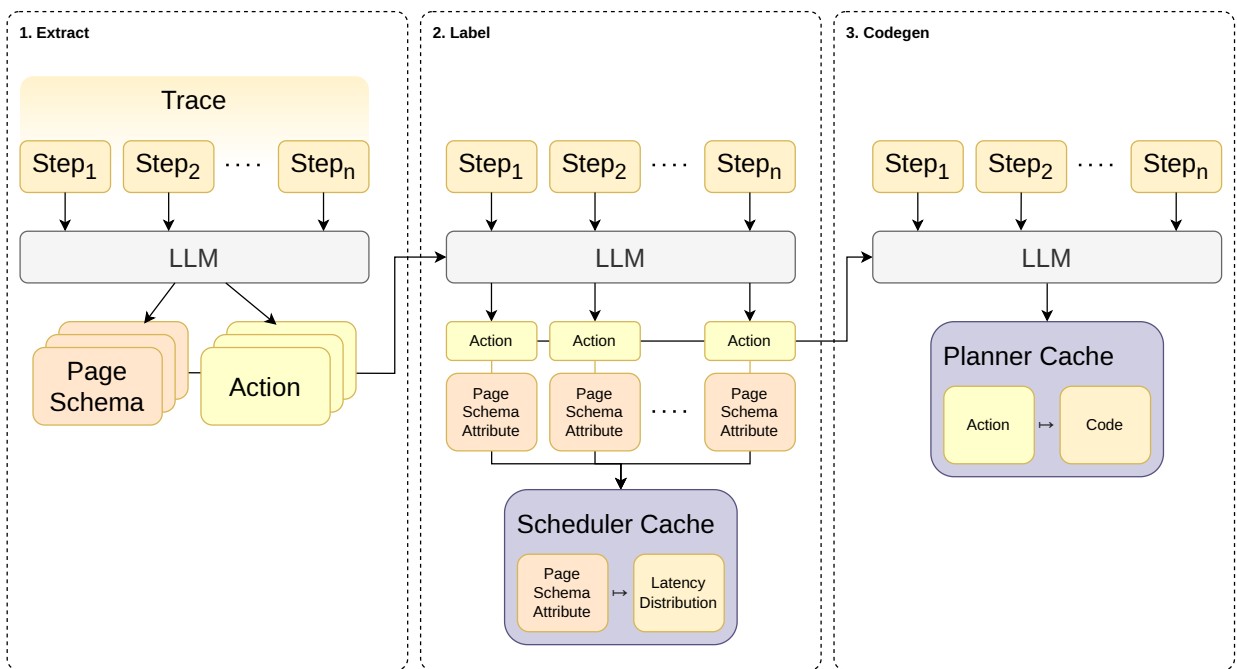

*Figure 11.* **Offline Cache Update.** Execution traces are processed to generate page schemas, map actions to schema elements, fit latency distributions, and update both planner cache (action $\mapsto$ code) and scheduler cache (element $\mapsto$ distribution). This process is similar to tool discovery approaches explored in prior work (Prabhu et al., 2025), though we focus on the protocol enforcement and cost-optimizing planning and scheduling aspects.

## B. All Planner Results

We provide complete latency and accuracy results across all five methods and applications. Table 2 reports end-to-end latency in seconds, and Table 3 reports task completion accuracy.

### B.1. CUA +cache Ablation

To isolate the contribution of cost-optimizing planning from tool access (Section 5), we give GPT-5.4 access to the same synthesized tools used by Browser-Use +cache and JIT-Planner, on the three REAL applications (Dashdish, Gomail, Omnizon; 27 tasks, 3 trials each, 81 runs). Table 4 reports per-application latency and accuracy against the relevant baselines; Table 5 reports the head-to-head against JIT-Planner across cardinality and length tiers, with 95% bootstrap confidence intervals.

Even with identical tools, CUA +cache remains 1.5–2.4× slower than JIT-Planner at comparable accuracy across all tiers, despite GPT-5.4 being a more capable, computer-use-native model than GPT-4.1. This indicates that the speedup is attributable to cost-optimizing planning rather than tool access alone.

## C. Statistical Significance

Here we report 95% confidence intervals and significance tests for the paper's main claims. We use paired Wilcoxon signed-rank tests and paired bootstrap confidence intervals for latency comparisons, and two-proportion $z$-tests with Wilson score intervals for accuracy comparisons.

### C.1. JIT-Planner (Table 1)

Latency claims use paired bootstrap confidence intervals on the ratio of mean latencies (10,000 resamples across the 111 task–model configurations), with paired Wilcoxon signed-rank tests for significance; the accuracy claim uses a two-proportion $z$-test. All headline JIT-Planner claims are highly significant.

*Table 6.* **JIT-Planner significance.** Latency rows: bootstrap 95% CI on the ratio of mean latencies, Wilcoxon $p$. Accuracy row: two-proportion $z$-test. Final latency row: per-task paired worst-/best-cost latency ratio.

| Claim | 95% CI | $p$ |
|---|---|---|
| $10.4\times$ speedup vs Browser-Use | $[9.0\times, 12.8\times]$ | 7.2e-20 |
| $6.8\times$ speedup vs Browser-Use +cache | $[5.8\times, 8.1\times]$ | 6.0e-20 |
| $1.8\times$ worst- vs. best-cost plan | $[1.6\times, 2.1\times]$ | 4.6e-12 |
| $+28$ pp accuracy vs Browser-Use | $[+19, +34]$ pp | 3.3e-08 |

The third row reports cost-aware plan selection: the per-task paired ratio between worst- and best-cost plans. The $5.3\times$ in Section 5 is the corresponding ratio of *mean* latencies (11.7s vs. 61.7s), which a few high-variance tasks inflate.

### C.2. JIT-Scheduler (Scheduler RQ2–RQ3)

Over the 27 REAL tasks per model, paired Wilcoxon tests show JIT-Scheduler is significantly faster than OpenAI CUA across all three models (Table 7). Its accuracy improvements over fixed strategies are also significant for Gemini-2.5-Pro (vs. both Serial and Hedge) and for GPT-4.1 vs. Serial (Table 8).

*Table 7.* **JIT-Scheduler vs. OpenAI CUA.** Over the 27 REAL tasks per model. Speedup is the ratio of mean latencies with bootstrap 95% CI; $p$ from paired Wilcoxon.

| Model | JIT-Scheduler | OpenAI CUA | Speedup [95% CI] ($p$) |
|---|---|---|---|
| GPT-4.1 | 142.6s, 77.8% | 258.7s, 77.8% | $1.8\times$ $[1.3, 2.5]$ (1.7e-03) |
| Gemini-2.5-Flash | 118.4s, 60.5% | 258.7s, 77.8% | $2.2\times$ $[1.4, 3.4]$ (2.7e-03) |
| Gemini-2.5-Pro | 109.9s, 86.4% | 258.7s, 77.8% | $2.4\times$ $[1.8, 3.1]$ (4.1e-05) |

*Table 8.* **JIT-Scheduler accuracy gain over fixed strategies**, in percentage points. Differences use a two-proportion $z$-test with Wilson 95% CIs; bold indicates $p < 0.05$.

| Model | vs Serial: $\Delta$ [95% CI] ($p$) | vs Hedge: $\Delta$ [95% CI] ($p$) |
|---|---|---|
| GPT-4.1 | $+\textbf{19.8}$ $[+8.6, +33.3]$ **(0.004)** | $+7.4$ $[-2.5, +18.5]$ (0.40) |
| Gemini-2.5-Flash | $+4.9$ $[-3.7, +14.8]$ (0.29) | $+4.9$ $[-6.2, +16.0]$ (0.42) |
| Gemini-2.5-Pro | $+\textbf{16.0}$ $[+6.2, +28.4]$ **(0.016)** | $+\textbf{14.8}$ $[+4.9, +24.7]$ **(0.006)** |

JIT-Scheduler attains significantly higher accuracy than Hedge for Gemini-2.5-Pro because it selects Parallel for suitable tasks, decomposing them into individually more reliable subtasks; it selects Hedge for only 15–22% of tasks. For the other models the accuracy differences are not significant; JIT-Scheduler's primary advantage is that it requires no manual, per-task strategy selection.

### C.3. Protocol Enforcement (Planner RQ1)

Two-proportion $z$-tests with Wilson score intervals ($n = 1{,}184$ trials per condition per model, across all plan-generation tasks and candidate samples) confirm that protocol enforcement significantly improves the valid-plan rate for every model.

*Table 9.* **Protocol enforcement effect on valid-plan rate.** Two-proportion $z$-test with Wilson 95% CI.

| Model | With protocol | Without protocol | $\Delta$ ($p$) |
|---|---|---|---|
| GPT-4.1 | 91.0% $[89.2, 92.5]$ | 78.0% $[75.5, 80.2]$ | $+13.0$ pp ($2.4 \times 10^{-18}$) |
| Gemini-2.5-Flash | 85.0% $[82.8, 86.9]$ | 74.2% $[71.7, 76.6]$ | $+10.7$ pp ($9.4 \times 10^{-11}$) |
| Gemini-2.5-Pro | 95.9% $[94.7, 96.9]$ | 79.1% $[76.7, 81.4]$ | $+16.8$ pp ($3.2 \times 10^{-35}$) |

All three models show a significant effect ($p < 10^{-10}$). Aggregated across models, the valid-plan rate improves from 77.1% to 90.6% ($+13.5$ pp; $z = 15.48$, $p = 4.5 \times 10^{-54}$).

## D. Task Benchmark and Oracle Strategy Analysis

**Task selection.** For each of the three REAL applications (Dashdish, Gomail, Omnizon), we select three tasks from the benchmark and manually curate six additional tasks. The nine tasks per application are designed so that three are latency-optimal under each scheduling strategy (Serial, Parallel, Hedge), ensuring balanced coverage for evaluating the scheduler. For WebArena (GitLab, Reddit), we randomly select five tasks per application. Because WebArena tasks are primarily sequential, they are evaluated with the planner only and do not have oracle scheduling strategies assigned.

Table 10 presents all 37 tasks with their latency-optimal scheduling strategies. Notably, high cardinality does not imply that parallelization is fastest. For example, "how many restaurants in the 'Light & fresh' category offer delivery?" (cardinality 10) is optimal under Serial, and "clear all my emails" (cardinality 50) is under Hedge. By contrast, the lower-cardinality task, "which of Winstop, Man vs. Fries, McDonald's, Popeyes, and Ike's has the cheapest fries?" (cardinality 5) is optimal under Parallel. This motivates our dynamic cost-aware scheduling over heuristics based on cardinality alone.

## E. Tool Protocol Example

Section 3.1 shows an example of a protocol-compliant tool. `add_to_cart` is a tool from the Dashdish application, which adds a menu item to the user's cart from a restaurant store page. `pre_tools` is an optional field that specifies the source tools for each input parameter.

*Listing 1.* Protocol-compliant tool manifest for `add_to_cart` (Dashdish). The `execute` body is abbreviated; the full implementation performs fuzzy item matching, modal interaction, customization application, and cart button clicking via DOM APIs.

```
1  {
2    "name": "add_to_cart",
3    "description": "Adds a specific item to the shopping cart with optional
4                   customizations and special instructions.",
5    "type": "setFields",
6    "input_schema": {
7      "type": "object",
8      "properties": {
9        "item_name": { "type": "string",
10                  "description": "The name of the item to add." },
11       "customizations": { "type": "string",
12                    "description": "Comma-separated 'Key: Value' pairs
13                    (e.g., 'Size: Large, Special Instructions: No onions')." }
14     },
15     "required": ["item_name"]
16   },
17   "output_schema": {
18     "type": "object",
19     "properties": {
20       "success": { "type": "boolean" },
21       "item_added": { "type": "string" },
22       "cart_count": { "type": "integer" },
23       "error": { "type": "string" }
24     },
25     "required": ["success"]
26   },
27   "pre": { "page_type": "store" },
28   "post": { "page_type": "store" },
29   "pre_check": "return document.body.textContent.includes('Full Menu')
30             ? true : [false, 'Not on a store page'];",
31   "post_check": "return output.success === true
32              ? true : [false, output.error];",
33   "execute": "// 1. Fuzzy-match item_name against menu cards
34            // 2. Click item card to open customization modal
```

```
35              // 3. Apply customizations (size, special instructions)
36              // 4. Click 'Add to Cart' button in modal
37              // 5. Return { success, item_added, cart_count }
38              // (Full implementation is around 50 lines)",
39     "pre_tools": {
40       "item_name": ["list_menu_items"],
41       "customizations": ["get_item_details"]
42     }
43   }
```

Below, we discuss the key fields that enable compositional verification:

- `pre`/`post`: The tool requires `page_type = "store"` (precondition) and guarantees the page remains on the store after execution (postcondition). The planner verifies that any plan calling `add_to_cart` first navigates to a store page.
- `pre_check`/`post_check`: Runtime DOM predicates verify the "Full Menu" heading is visible before execution and that the operation succeeded after execution, enabling debugging of state flow violations.
- `input_schema`/`output_schema`: Full JSON Schema type constraints enable static type checking of tool call arguments and return values in generated plans.
- `pre_tools`: Declares that `item_name` should be chosen from the output of `list_menu_items` and `customizations` from `get_item_details`.

## F. Planner Code Plan Examples

We illustrate the planner's candidate generation and cost-based selection (Algorithm 1) with code plans for task `dashdish-custom-6`: *"Visit the first five restaurants in 'Best near you' and report how many of them have under 20 reviews."* The planner generates $k$ candidates in parallel; below we show three representative plans generated by Gemini-2.0-Flash-Lite from actual experimental runs.

### Plan A: Rejected (State-Flow Violation)

*Listing 2.* Plan A — rejected due to state-flow violation at line 4.

```
1  stores = await list_all_stores(detailed=True)
2  best_near_you_stores = stores.items[:5]
3  under_20_reviews_count = 0
4  for store in best_near_you_stores:
5      store_details = await get_store_details() # FAILS
6      if store_details.review_count < 20:
7          under_20_reviews_count += 1
8  result = await ai_eval("...", count=under_20_reviews_count)
```

**Explanation.** `get_store_details` requires `pre.page_type = "store"`, but `list_all_stores` keeps the state at `page_type = "home"`. The code never calls `goto_store` before `get_store_details`, so the precondition $s \not\supseteq \mathcal{M}[\text{get\_store\_details}].\text{pre}$ is violated (Algorithm 1, line 12) and the plan is rejected.

### Plan B: Accepted but Suboptimal (Unnecessary `ai_eval`)

*Listing 3.* Plan B — valid but suboptimal due to unnecessary LLM call.

```
1  stores = await list_all_stores(detailed=True)
2  best_near_you = [s for s in stores.items
3                if s.rating is not None]
4  first_five = best_near_you[:5]
5  count = 0
6  for store in first_five:
7      if store.review_count is not None
8          and store.review_count < 20:
9          count += 1
10 result = await ai_eval( # unnecessary
11     "The_number_of_restaurants_with_under_20_"
12     "reviews_is_{count}.", count=count)
```

**CFG cost** (Algorithm 1, lines 9–18): One tool call and one `ai_eval` call, both at loop depth $d = 0$:

$$\text{cost}_B = C_{\text{tool}} \cdot \gamma^0 + C_{\text{eval}} \cdot \gamma^0 = 0.1 + 10.0 = 10.10$$

The `ai_eval` dominates the cost, but it is entirely unnecessary, as the count can be computed in pure code.

**Plan C: Latency-Optimal (Selected)**

*Listing 4.* Plan C — latency-optimal; pure code, no LLM calls.

```
1   await goto_home()
2   stores = await list_all_stores(detailed=True)
3   first_five = stores.items[:5]
4   count = 0
5   for store in first_five:
6       if store.review_count is not None
7           and store.review_count < 20:
8               count += 1
9   result = f"Out␣of␣the␣first␣five␣restaurants,␣" \
10          f"{count}␣have␣under␣20␣reviews."
```

**CFG cost:** Two tool calls at depth $d = 0$, no `ai_eval`:

$$\text{cost}_C = 2 \cdot C_{\text{tool}} \cdot \gamma^0 = 2 \times 0.1 = 0.20$$

**Comparison:** Plan A is rejected (invalid state flow). Plan B is valid but incurs an unnecessary `ai_eval` call that dominates cost (10.10 vs. 0.20). Plan C achieves approximately a $50\times$ cost reduction by formatting the result in pure code. The planner therefore selects Plan C via $\text{argmin}$ (Algorithm 1, line 25).

# G. Scheduler Element Usage Example

We illustrate the scheduler's Monte Carlo cost estimation (Algorithm 2) with two examples that demonstrate when each strategy is preferred.

**Example 1: Low-Variance Elements Favor Serial**

Task `dashdish-custom-1`: *"How much is a matcha croissant and a regular iced matcha latte from Stonemill Matcha? What about a ceremonial matcha latte and a mochi set?"* (cardinality 4, length 2).

**Step 1: LLM-Predicted Element Usage.**

The scheduler first predicts sequential element usage (Algorithm 2, line 5). The LLM reasons that the task requires navigating to the Stonemill Matcha store page (1 click on `restaurantCard`), after which all four item prices are *readable* from the visible `fullMenuItemCard` properties without further interaction:

| Element | Page | Action | Count |
|---|---|---|---|
| `restaurantCard` | main → store | CLICK | 1 |
| `fullMenuItemCard` | store | READ | 0 |

The parallelizability check returns **True** (4 independent item lookups), producing a parallel plan with 4 workers, each navigating via `restaurantCard` $\times 1$. The hedge strategy runs $n = 4$ replicas of the full serial plan and takes the first to finish.

**Step 2: Monte Carlo Latency Estimation.**

The scheduler samples from the cached latency distribution for `restaurantCard`, fitted from 20 observed traces: Weibull($k$=3.60, $\lambda$=10.25) with mean 9.32 s and $\sigma$=2.37 s.

**Implementation note.** In practice, we augment the core cost model (Algorithm 2) with two additions that improve estimation accuracy:

- **Page read cost** ($C_{\text{read}} = 5$–$10$ s, model-dependent): the sampled element latency captures the cost of *finding and interacting* with an element on a page, but navigating to a new page incurs additional cost (loading, screenshot capture, DOM extraction). We add a fixed $(1 + \text{num\_navigations}) \times C_{\text{read}}$ per trial.
- **Repeat interaction discount** ($C_{\text{repeat}} = 5$–$7$ s, model-dependent): when an element is interacted with multiple times, only the first interaction is sampled from the fitted distribution. This represents the cost of *discovering* the element.

Subsequent interactions use a fixed $C_{\text{repeat}} < \mathbb{E}[\mathcal{D}_e]$, since the agent already knows where the element is and does not need to re-discover it.

Below we show one representative trial out of $N_{\text{MC}}$=1000, followed by the aggregated results. Let $R = (1 + \text{num\_navigations}) \times C_{\text{read}}$ denote the page read cost (here $R = 2 \times 5 = 10\,\text{s}$).

**Serial.** One worker executes the full plan: sample one `restaurantCard` latency, add $R$:

$$\ell_{\text{ser}} = \underbrace{8.4}_{\text{sample}} + \underbrace{10}_{R} = 18.4\,\text{s}$$

**Hedge.** Four replicas race on the same serial plan; take the fastest plus overhead $\delta_h$:

$$\ell_{\text{hed}} = \min(\underbrace{18.4, 19.1, 17.6, 20.2}_{\text{4 independent serial samples}}) + \delta_h = 17.6 + 5 = 22.6\,\text{s}$$

**Parallel.** Each of the 4 workers handles one item and navigates once (`restaurantCard` $\times 1$); because workers do not share a session, each incurs its own navigation, so its latency is a single `restaurantCard` draw plus the page read cost $R$. The batch latency is the slowest worker plus overhead $\delta_p$:

$$w_1 = 8.4 + R = 18.4\,\text{s}$$
$$w_2 = 11.5 + R = 21.5\,\text{s}$$
$$w_3 = 7.2 + R = 17.2\,\text{s}$$
$$w_4 = 9.8 + R = 19.8\,\text{s}$$
$$\ell_{\text{par}} = \max(18.4, 21.5, 17.2, 19.8) + \delta_p = 21.5 + 20 = 41.5\,\text{s}$$

**Explanation.** On this trial, Serial (18.4 s) is fastest, followed by Hedge (22.6 s) and then Parallel (41.5 s); aggregated over all $N_{\text{MC}} = 1000$ trials, Serial has the lowest mean latency and the highest win rate, and the scheduler selects it. Because the four prices are readable from a single page after one navigation, there is little to parallelize: each parallel worker must still perform its own navigation, so spawning 4 workers adds coordination overhead $\delta_p$ (20 s) without reducing the core work. Hedge benefits from taking the $\min$ over 4 replicas, which lowers the navigation latency, but its $\delta_h$ overhead still leaves it slower than Serial for this single-navigation task.

### Example 2: High-Variance Elements Favor Hedging

Task `v2/dashdish-10`: *"Place an order for any type of sub-sandwich, keep the total under $30."* (cardinality 1, length 6). The sequential plan requires 6 element interactions across 4 pages:

| Element | Navigation | Count | Mean | $\sigma$ |
|---|---|---|---|---|
| `restaurantCard` | main $\rightarrow$ store | 1 | 9.3 s | 2.4 s |
| `fullMenuItemAddButton` | $\rightarrow$ modal | 1 | 24.8 s | **25.8 s** |
| `modal.addToCartButton` | — | 1 | 9.0 s | 0 |
| `cartIconButton` | $\rightarrow$ modal | 1 | 35.8 s | **12.4 s** |
| `modal.checkoutButton` | store $\rightarrow$ checkout | 1 | 14.9 s | 6.5 s |
| `placeOrderButton` | $\rightarrow$ confirmation | 1 | 10.4 s | 0 |

The parallelizability check returns False, so only Serial and Hedge are compared. The key element in the plan is `fullMenuItemAddButton`: it has a Gamma distribution ($k$=1.31, $\theta$=18.95) with extreme variance ($\sigma = 25.8\,\text{s}$, range 6–91 s), reflecting that locating the correct add button on a menu page is highly variable. Hedging with $n$=4 replicas takes $\min$ over 4 independent draws from this heavy-tailed distribution, substantially reducing the expected latency of this bottleneck step.

| | **Serial** | **Hedge (4 replicas)** |
|---|---|---|
| $\bar{L}_\sigma$ | 133.50 s | **114.71 s** |
| Win rate | 27.4% | **72.6%** |

The scheduler correctly selects **Hedge**: the $\min$ over replicas tames the heavy tail of the high-variance elements, more than offsetting the $\delta_h$ overhead. This contrasts with Example 1, where low variance made the overhead not worthwhile.

# H. Experimental Setup

### Models

We use GPT-4.1 for all offline cache stages (Extract, Label, Codegen; Section A). For online experiments, each run uses a single model consistently for plan generation and scheduling. The set of models varies by experiment:

- **Code generation** (Figures 5 and 8): GPT-OSS-120B, GPT-4.1, GPT-5-mini, GPT-5, Gemini-2.0-Flash-Lite, Gemini-2.5-Flash, Gemini-2.5-Pro.
- **End-to-end planner and scheduler** (Figures 6 and 10): GPT-4.1, GPT-5.4, Gemini-2.5-Flash, Gemini-2.5-Pro, Anthropic CUA (`claude-sonnet-4-20250514`), OpenAI CUA (`computer-use-preview-2025-03-11`).

### Infrastructure

- **Compute:** AWS `m8i.xlarge` instance (4 vCPUs, 16 GiB RAM)
- **Browser automation:** Browser-Use v0.7.10 (Yang et al., 2024) wrapped by BLAST[1] for parallelism and request hedging, using Chromium via raw Chrome DevTools Protocol (CDP)
- **Parallelism:** Each worker runs an isolated browser context; up to $n = 4$ workers for scheduling strategies
- **Benchmarks:** REAL benchmark tasks run on officially hosted Vercel deployments (Dashdish, Omnizon, Gomail); WebArena tasks run on a self-hosted WebArena environment

### Hyperparameters

| Symbol | Value | Description |
|---|---|---|
| *Planner (Algorithm 1)* | | |
| $k$ | 32 | Number of valid candidate plans before selection |
| $M_{\max}$ | 1 | Maximum planner iterations per worker |
| $C_{\text{tool}}$ | 0.1 | Base cost weight for tool calls |
| $C_{\text{eval}}$ | 10.0 | Base cost weight for `ai_eval` calls |
| $\gamma$ | 10 | Loop nesting penalty (cost decay factor) |
| *Scheduler (Algorithm 2)* | | |
| $N_{\text{MC}}$ | 1000 | Number of Monte Carlo trials for latency estimation |
| $n$ | 4 | Number of parallel workers (vCPU budget) |
| $\delta_p$ | 20–30s | Parallel strategy overhead; model-dependent |
| $\delta_h$ | 5–15s | Hedge strategy overhead; model-dependent |

### Evaluation Protocol

- Each task is evaluated over **3 independent runs**; we report the mean accuracy and mean latency.
- **Accuracy** is measured via the benchmark harness: REAL uses state-diff checking; WebArena uses pre-defined evaluation functions.
- **Latency** is measured as wall-clock time from task submission to completion, including planning overhead.
- Oracle scheduling strategies are determined by running all three strategies (Serial, Parallel, Hedge) on each task with GPT-4.1 and selecting the one with minimum mean latency across 3 runs.

### Cost Model

The planner's cost model groups execution steps into three tiers with different latencies: cached tool calls (∼0.5–2s), `ai_eval` (LLM) calls (∼5–15s), and agentic sub-loops (∼30–60s+). Although latencies vary within each tier, the variance *within* tiers is small relative to the variance *across* tiers, so the model's role is to distinguish plans by the number of LLM calls and loop-nesting depth rather than to predict absolute latency. Accordingly, the estimate is used only to *rank* candidates: best-cost selection achieves $1.8\times$ $[1.6\times, 2.1\times]$ lower latency than worst-cost selection (paired Wilcoxon $p = 4.6\text{e-}12$; Section C), confirming that the model ranks plans effectively despite its simplicity.

---

[1] https://github.com/stanford-mast/blast, commit `1ded566`.

# I. Prompt Templates

We provide the prompt templates used in the system, organized by pipeline stage. Placeholders marked with <<...>> are filled at runtime. **Online prompts** (Prompts 1–2) are invoked at task time for planning and scheduling. **Offline prompts** (Prompts 3–5) are invoked during the cache population pipeline (Section A, Figure 11), corresponding to the three stages: Extract, Label, and Codegen.

## Online Prompts

### Prompt 1: Plan Generation (Algorithm 1, line 7)

This prompt is used by each planner worker to generate a code plan from a task description and tool manifests. It consists of a **system message** (fixed instructions and few-shot examples) and a **user message** (tool definitions, state, rules, and task). On retry after a validation error, the user message is replaced with the error string to enable iterative refinement.

*Listing 5.* Plan generation prompt — system message.

```
You are an expert in writing code that calls tools and programmatically
asks AI questions.

<examples>
Here are examples of good code to generate. Use them as a reference, but
do not copy them verbatim.

E1: immediate return
 result = "The response to the user's task can be immediately returned"

E2: call single tool with handling of empty tool result
 x = await tool_a(42)
 result = f"It is {x.items[0].field1}" if x.items else "No results"

E3: result with ai_eval
 x = await tool_b(param="value")
 result = await ai_eval("Summary of {data}", data=x.content)

E4: ordering tool calls based on the current STATE and tool
   preconditions/postconditions # <-- state-aware label
 await tool_c()
 items = await tool_d()
 collected = {}
 for item in items.items:
    await tool_e(name=item.name)
    data = await tool_f()
    collected[item.name] = data.info
 result = collected

E5: multiple tool calls with ai_eval for matching user terms
 await tool_c()
 x = await tool_d(id=123)
 y = await tool_e(p1=x.data)
 closest = await ai_eval("Name in {opts} closest to 'the titanic'",
                  opts=y.options)
 z = await tool_f(item_name=closest)
 result = await ai_eval("Response about {info}", info=z.info)

E6: control flow with loops and conditionals
 await tool_c()
 x = await tool_e(123)
 results = [await tool_f(item_name=i.name) for i in x.items if i.value>3]
 result = await ai_eval("Summary of {results}", results=results)

E7: use information from previous tool calls to filter dynamically
 tools = await list_items()
 tools = [t for t in tools if "keyword" in t.name.lower()]
 details = await get_tool_details(tool_id=tools[0].id)
 result = details.info

E8: use contains matching instead of exact key lookup
 items = await list_items()
 matching = [i for i in items if "target" in i.name.lower()]
 price = matching[0].price if matching else None
</examples>

<instructions>
Generate completion of the given code block to implement the TASK.
If an error is reported, fix the previously generated code accordingly.
```

```
</instructions>
```

*Listing 6.* Plan generation prompt — user message. Tool stubs are auto-generated from manifests (Section 3.1); the model completes the code after the final comment.

```
Now is <<timestamp>>

<<agent_description>> # if provided

```python
from pydantic import BaseModel, Field
from typing import Optional, List, Dict, Any

# --- Auto-generated from tool manifests ---
STATE = {<<keys_from_all_tool_pre_post>>}
TOOLS_CALLED: set[str] = set()

<<For each tool: input/output Pydantic models, async function stub
  with typed params, docstring, and contract assertions:
    assert STATE[k] == v for each (k,v) in tool.pre # precondition
    ... # implementation
    STATE[k] = v for each (k,v) in tool.post # postcondition
    TOOLS_CALLED.add("tool_name")>>

async def ai_eval(expr: str, **kwargs) -> str: ...
async def ask_human(question: str) -> str: ...

# INITIAL STATE. DO NOT ACCESS OR MODIFY.
STATE.update(<<initial_state>>) # if state-aware

# RULES:
# - Do not generate comments
# - If using a user-provided term, use ai_eval to determine
# which option it most closely matches.
# - When calling ai_eval, always specify (1) output format,
# (2) how descriptive the response should be.
# - Generate code that directly executes the task.
# - Use ask_human if no tool's precondition is satisfied.
# - Do not access or modify STATE directly. Do not call get_url()

# TASK: Create a response for the following input from the user:
# <<task_description>>
#
# RESPOND WITH ONLY YOUR CODE AFTER THIS USING ABOVE DEFINITIONS
```
```

### Prompt 2: Element Usage Prediction (Algorithm 2, line 5)

The scheduler predicts element usage via a three-step prompting pipeline: (A) sequential element usage prediction, (B) parallelizability classification, and (C) parallel element usage prediction. Step C is only invoked if Step B returns true. Serial and hedge latency estimates use Step A output; parallel estimates additionally use Step C output.

*Listing 7.* Step A — Sequential element usage prediction.

```
SYSTEM:
You are a helpful assistant who predicts element usage for web
automation tasks. Given a task, page schemas, and available elements,
predict which elements will be used.

Visibility Hierarchy:
Elements are organized into pages and modals. Information becomes
visible as you navigate:
1. Page elements: navigating to a page (via opensPage) makes all
   non-modal elements visible. Properties can be READ without clicking.
2. Modal elements: only visible AFTER clicking an opensModal element.

Key principle: READ = be on the right page. ACCESS content behind
opensPage/opensModal = INTERACT (click).

Example visibility chain:
  [Page: main] -> itemCard visible (READ itemCard.name, .price)
    click itemCard (opensPage: "detail")
  [Page: detail] -> detailButton visible, reviewCard visible
    click detailButton (opensModal: "editModal")
  [Modal: editModal] -> editModal.saveButton now visible
```

```
Counting rules:
- Only count elements that MUST be interacted with (clicked, typed)
- READ-only = count 0; INTERACT = count >= 1
- Use x-visibleCount / x-visibleCounts for accurate counts
- Never output count=0 for required interaction elements

Examples:
- "What is the price of the first product?"
  productCard has {name, price}, opensPage -> count: 0 (just READ)
- "Get details from the first product's detail page"
  productCard has opensPage -> count: 1 (must click to navigate)
- "Add 2 items to cart" (requires customization modal)
  itemCard -> 2, customizationModal.addButton -> 2

Produce the simplest plan with the minimum interactions needed.
ONLY use elements from the provided list.

USER:
Task: <<task_description>>

<<page_schemas>>
Available Elements (grouped by page/modal):
<<valid_elements>>

Property Reference:
- Properties without opensPage/opensModal: READ once visible
- opensPage: click navigates to another page
- opensModal: click opens modal
- x-visibleCount: total visible instances
- x-visibleCounts: count per group (e.g., {"SectionA": 5})

Output a SchedulerPlan with:
- elements: [{element_name, required (bool), count (int)}]
```

*Listing 8.* Step B — Parallelizability check.

```
SYSTEM:
Determine if a task can be parallelized.
Parallelizable = same action repeated on multiple items/entities
  (e.g., writing reviews for multiple orders, visiting multiple
  restaurants, sending emails to multiple people).
NOT parallelizable = single entity operation, purely sequential
  dependencies, or single-page data extraction.

USER:
Task: <<task_description>>
Is this parallelizable?
(True if it involves multiple independent operations on separate entities)
```

*Listing 9.* Step C — Parallel element usage prediction (invoked only if Step B returns true).

```
SYSTEM:
Identify parallelizable operations from a sequential plan.
All elements in the list are REQUIRED for the task.

IMPORTANT: Maximum 4 workers can run simultaneously. If more than
4 parallel operations are needed, and workers are batched
(e.g., 6 workers = batch of 4, then batch of 2).

Output:
- parallel_elements: elements that run independently (per worker)
- num_workers: total parallel workers needed (can exceed 4)
- requires_rediscovery: true if workers must re-navigate to find
  their target (element has no URL/link property in schema)
- rediscovery_elements: navigation elements each worker needs
  if requires_rediscovery = true

USER:
Task: <<task_description>>
Required Elements (all must be executed): <<sequential_usage_from_step_A>>

Identify parallel operations. If the element is used N times independently,
set num_workers=N and count=1 per worker.
```

**Offline Prompts (Figure 11: Extract → Label → Codegen)**

**Prompt 3: Schema Extraction (Figure 11, Stage 1 — Extract)**

This prompt maps page screenshots and optional HTML snapshots to a JSON page schema (JSON Schema Draft 2020-12). The schema defines actionable elements as `$defs`—each representing a discrete user action (click, type, toggle)—with properties for visible data, `opensPage`/`opensModal` for transitions, and `x-visibleCount`/`x-visibleCounts` for element cardinality. The output populates the page schemas used by both the planner and scheduler caches.

*Listing 10.* Schema extraction prompt — maps page screenshots to a JSON page schema.

```
Produce a JSON schema (Draft 2020-12) for this webpage.

Important Rules:
1. Only describe what is visible in the screenshot(s) or HTML snapshot(s).
  Do not generate or infer properties that are not explicitly visible.

2. Distinction between Elements and Properties:
  - $defs = ACTIONABLE ELEMENTS ONLY (click, type, toggle)
    Examples: buttons, clickable cards, input fields, links
  - Properties within $defs = INFORMATION visible on that element
    Examples: restaurantName, price, rating are properties OF
    restaurantCard, not separate elements

3. Structure: Keep schema FLAT -- one $def per actionable element type.
  Do not create $defs for containers/wrappers.

4. Modal Rules:
  - Modals include popups, sidebars, drawers, dialogs, overlays
  - Modal $def names MUST include "Modal" (e.g., "cartSidebarModal")
  - Elements inside modals use dot notation: "cartModal.checkoutButton"
  - Trigger elements MUST have opensModal property

5. Page Navigation:
  - Elements navigating to a different page MUST have opensPage
  - opensPage value must be a known page name for this site
  - opensModal is for overlays on the same page (no URL change)

6. Consolidate Similar Elements:
  - Create ONE generic $def with a grouping property, not separate
    $defs per section
  - Use x-visibleCounts for per-group breakdown

7. Cardinality:
  - Add x-visibleCount for repeated elements (total visible count)
  - If grouped, add x-visibleCounts with per-group breakdown

Example of a well-structured $def:
  "itemCard": {
    "type": "object",
    "description": "Clickable card navigating to item details",
    "x-visibleCount": 27,
    "x-visibleCounts": {"Section A": 4, "Section B": 3, ...},
    "properties": {
      "name": {"type": "string"},
      "price": {"type": "string"},
      "opensPage": {"const": "detail"}
    }
  }

URL pattern: <<url_pattern>>
<<known_pages_section>>
<<existing_defs_section>>
<<html_section>>
<<modal_hint_section>>
Find attached the screenshot(s):
```

**Prompt 4: Action-to-Element Labeling (Figure 11, Stage 2 — Label)**

This prompt maps each step in an execution trace to a page schema element, identifying the agent's *ultimate* interaction target (not intermediate exploratory actions like scrolling). The resulting element-to-step mappings, combined with step timestamps, are used to fit per-element latency distributions for the scheduler cache.

*Listing 11.* Action labeling prompt — maps trace steps to page schema elements for the scheduler cache.

```
SYSTEM:
You are an expert at analyzing the actions of an AI agent.
Given an agent step, fill out the tuple:
  (id: int, goal_element: str, success: bool,
   is_modal: bool, modal_name: str)

Elements = User Actions:
Each element in valid_elements represents a DISCRETE USER ACTION.
Elements are NOT data properties. For example:
- "restaurantCard" = clicking a restaurant card (navigates to menu)
- "addToCartButton" = clicking to add item to cart
- "searchInput" = typing in the search field

DO NOT use property-like names (e.g., "restaurantCard.restaurantName").
These are informational properties, NOT separate actions.

Intermediate vs Ultimate Goal:
Many steps are EXPLORATORY (scrolling, extracting data, clicking to
reveal elements). Map to the ULTIMATE target, not the intermediate
action.

Examples:
1. Agent says "scroll to find the submit button" + scroll action:
   WRONG: goal_element = "scrollable container"
   CORRECT: goal_element = "submitButton" (the ultimate target)

2. Agent says "click restaurant card to go to menu page":
   WRONG: goal_element = "restaurantCard.destinationUrl"
   CORRECT: goal_element = "restaurantCard" (the action is the click)

3. Agent says "extract data to check for add-to-cart buttons":
   CORRECT: goal_element = "addToCartButton" (what they seek)

Guidelines:
- goal_element MUST be from valid_elements list
- Use parent element name, not property-like names
- If goal_element is from a modal, set is_modal=True + modal_name
- If exploratory action, identify what element they are trying to
  FIND or REACH

USER:
<valid_elements>
<<valid_elements>>
</valid_elements>

<trace>
<<trace_steps_with_goal_and_thinking>>
</trace>
```

### Prompt 5: Tool Synthesis (Figure 11, Stage 3 — Codegen)

This prompt drives a synthesis agent that creates protocol-compliant tools, populating the planner cache (action $\mapsto$ code). The agent operates in a loop: it lists existing tools, observes browser state, and creates or updates tools as needed. Each tool is defined by JavaScript function bodies for `pre_check` (DOM readiness), `execute` (action), and `post_check` (completion verification), along with `pre`/`post` state dictionaries, matching the protocol fields in Section 3.1. A secondary sub-prompt (Listing 13) auto-generates `description`, `input_schema`, and `output_schema` from the execute script.

*Listing 12.* Tool synthesis agent prompt — main instruction.

```
Complete the TASK using ONLY protocol-compliant tool actions,
unless rewinding or requesting human assistance.

<workflow>
1. Call list_tools to see available tools
2. Observe
   a. If tool exists and pre matches current state: call it
   b. If tool missing or fails:
      i. If it exists, call list_tools with get_code_for=<tool_name>
      ii. Optionally call ask_html for DOM guidance on current page
      iii. Create/update tool with update_tool
      iv. Then call the new tool
3. If tool fails and partially progressed, rewind first
4. Loop until TASK complete
</workflow>

<javascript_requirements>
```

```
pre_check, execute, and post_check are FUNCTION BODIES only:
  CORRECT:
    pre_check: "return document.querySelector('#el') ? true
                : [false, 'Not loaded'];"
    execute: "return {items: [...]};"
    post_check: "return document.querySelector('h3')
                .textContent.includes(inputs.name)
                ? true : [false, 'Wrong page'];"
  WRONG:
    pre_check: "(function(){ })()"
    execute: "async function() { }"
</javascript_requirements>

<tool_types>
- observe: Returns state (page, selectedItem, etc.) -- one per domain
- listItems: Returns {items: [...]}
- getFields: Returns field values object
- setFilter: Sets a filter (search query, category) and applies it
- setFields: Edits fields (fill form) and/or proceeds (submit)
- gotoItem: Navigates to item (must work for any item from listItems)
- gotoField: Opens/focuses a field
</tool_types>

<update_tool_required>
name: Tool identifier (e.g. "list_restaurants")
type: One of the tool_types above
pre_check: JS body -- check DOM readiness before execute
execute: JS body -- perform the action (has access to 'inputs')
post_check: JS body -- verify DOM state after execute
pre: State dict before running (e.g. {"page": "list"})
post: State dict after running (e.g. {"page": "detail"})
input_parameters: Array of param names (e.g. ["restaurantName"]) or []
</update_tool_required>

<abstract_state>
State maps variable names to values or patterns:
- Concrete value: "home", 42, true
- "*": any non-null value
- "a|b|c": one of these values
- "$paramName": must match named input/output parameter
- "": must be null
</abstract_state>

<examples>
observe (no params):
 name: "observe_app", type: "observe", input_parameters: []
 pre_check: "return document.querySelector('#header') ? true
            : [false, 'Not loaded'];"
 execute: "const page = location.pathname === '/'
            ? 'home' : 'detail'; return {page};"
 pre: {}, post: {}

gotoItem (with params):
 name: "goto_product", type: "gotoItem"
 input_parameters: ["productName"]
 pre: {"page": "list"}
 post: {"page": "detail", "selectedProduct": "$productName"}
 pre_check: "return document.querySelector(
            '.product-link') ? true : [false, 'Not found'];"
 execute: "const link = Array.from(
            document.querySelectorAll('.product-link'))
            .find(el => el.textContent.includes(inputs.productName));
            if (link) link.click(); return {success: !!link};"
 post_check: "return document.querySelector('.detail h1')
             .textContent.includes(inputs.productName)
             ? true : [false, 'Detail not loaded'];"
</examples>

TASK:
<<task_or_goal_text>>
```

*Listing 13.* Schema generation sub-prompt — invoked inside update_tool to auto-generate description, input_schema, and output_schema from the execute script.

```
SYSTEM: You are a JSON generator. Output ONLY valid JSON.

USER:
Generate JSON with description, input_schema, and output_schema
for this tool.
```

```
Tool: <<tool_name>>
Type: <<tool_type>>
Input Parameters: <<input_parameters>>
Execute Script:
```javascript
<<execute_body>>
```

Respond with valid JSON only:
{
  "description": "What this tool does (one sentence)",
  "input_schema": {
    "type": "object", "properties": {}, "required": []
  },
  "output_schema": {
    "type": "object", "properties": {}, "required": []
  }
}

For input_schema: use parameters <<input_parameters>>.
Each needs type (string/number/boolean/array/object) and description.

For output_schema: analyze the execute script's return value.
Common patterns:
- listItems: {"items": {"type": "array", "items": {"type": "object",
          "properties": {...}, "required": [...]}}}
- observe: {"page": {"type": "string", "enum": [...]}, ...}
- setFields: {"success": {"type": "boolean"}}

CRITICAL: For array types, ALWAYS specify items schema with full detail.
```

*Table 2.* **End-to-end latency (seconds) across all methods and applications.**

| Application | Anthropic CUA | OpenAI CUA | Browser-Use | Browser-Use +cache | JIT-Planner |
|---|---|---|---|---|---|
| Dashdish | 137.7 | 277.0 | 118.8 | 62.8 | **14.5** |
| Gitlab | 166.4 | 133.1 | 144.7 | 69.7 | **9.9** |
| Gomail | 97.2 | 173.3 | 133.8 | 94.6 | **10.4** |
| Omnizon | 190.2 | 325.9 | 113.9 | 86.8 | **15.5** |
| Reddit | 121.3 | 475.4 | 81.3 | 82.7 | **4.3** |
| **Overall** | 142.6 | 276.9 | 118.5 | 79.3 | **10.9** |

*Table 3.* **Accuracy (%) across all methods and applications.**

| Application | Anthropic CUA | OpenAI CUA | Browser-Use | Browser-Use +cache | JIT-Planner |
|---|---|---|---|---|---|
| Dashdish | 85 | 85 | 65 | **100** | **100** |
| Gitlab | 47 | 80 | 49 | 67 | **87** |
| Gomail | 85 | 78 | 63 | 80 | **96** |
| Omnizon | 78 | 86 | 76 | 84 | **89** |
| Reddit | 67 | 47 | 72 | 89 | **93** |
| **Overall** | 72 | 75 | 65 | 84 | **93** |

*Table 4.* **CUA +cache ablation, per application.** Latency (s) and accuracy (%) on the three REAL applications. CUA +cache uses GPT-5.4; JIT-Planner and Browser-Use +cache use GPT-4.1. Lower latency is better.

| | Dashdish | | Gomail | | Omnizon | |
|---|---|---|---|---|---|---|
| Method | Lat. | Acc. | Lat. | Acc. | Lat. | Acc. |
| OpenAI CUA | 277.0 | 85 | 173.3 | 70 | 190.2 | 78 |
| Anthropic CUA | 137.7 | 83 | 97.2 | 85 | 113.9 | 76 |
| Browser-Use +cache | 62.8 | 100 | 94.6 | 80 | 86.8 | 84 |
| CUA +cache | 34.5 | 89 | 23.0 | 96 | 51.2 | 100 |
| JIT-Planner | **14.5** | 100 | **10.4** | 96 | **15.5** | 89 |

*Table 5.* **CUA +cache vs. JIT-Planner across complexity tiers.** Latency (s) with 95% bootstrap CIs; speedup of JIT-Planner over CUA +cache in parentheses. Accuracy (%) with Wilson 95% CIs.

| | JIT-Planner | | CUA +cache | |
|---|---|---|---|---|
| Tier | Lat. | Acc. | Lat. (speedup) | Acc. |
| C-Low | 15.8 [12.5, 19.2] | 90.0 | 23.4 [19.4, 27.6] (1.5×) | 100.0 |
| C-Med | 14.5 [11.4, 17.8] | 90.0 | 34.9 [26.3, 44.0] (2.4×) | 100.0 |
| C-High | 25.2 [14.0, 38.9] | 100.0 | 55.7 [38.6, 75.7] (2.2×) | 81.0 |
| T-Short | 10.4 [7.9, 13.1] | 100.0 | 15.3 [12.9, 18.2] (1.5×) | 90.3 |
| T-Med | 19.9 [17.0, 23.0] | 88.9 | 47.5 [39.6, 56.2] (2.4×) | 100.0 |
| T-Long | 24.5 [14.4, 36.4] | 87.5 | 49.4 [34.9, 66.5] (2.0×) | 95.8 |

*Table 10.* **Task Benchmark with Latency-Optimal Scheduling Strategies.** *Card.* = task cardinality (number of elements processed); *Len.* = task length (number of steps under baseline Browser-Use). Complexity tiers defined in Section 4: C-Low (= 1), C-Med (2–4), C-High (≥ 5); T-Short (1–5), T-Med (6–8), T-Long (≥ 9). Oracle schedules are determined using GPT-4.1, averaged over 3 runs. Task IDs prefixed with v1/ or v2/ denote tasks sourced from REAL benchmark versions 1 and 2, respectively; IDs of the form *-custom-* denote custom-curated tasks.

| Bench. | App | Task ID | Task Description | Card. | Len. | Oracle |
|---|---|---|---|---|---|---|
| REAL | Dashdish | dashdish-custom-1 | How much is a matcha croissant and a regular iced matcha latte from Stonemill Matcha? What about a ceremonial matcha latte and a mochi set? | 4 | 2 | Serial |
| | | dashdish-custom-2 | For each vendor under the "Costume party ready" category, visit the store page and get the first three reviews. | 9 | 8 | Parallel |
| | | dashdish-custom-3 | Buy a taco from any store and make sure you place the order. | 1 | 6 | Hedge |
| | | v2/dashdish-10 | Place an order for any type of sub-sandwich, keep the total under $30. | 1 | 6 | Hedge |
| | | v1/dashdish-1 | What are the first three restaurants listed on the homepage? | 3 | 1 | Serial |
| | | dashdish-custom-6 | Visit the first five restaurants in "Best near you" and report how many of them have under 20 reviews. | 5 | 8 | Parallel |
| | | dashdish-custom-7 | What's the most expensive item from Petco? | 1 | 5 | Hedge |
| | | v1/dashdish-3 | How many restaurants in the "Light & fresh" category offer delivery? | 10 | 2 | Serial |
| | | dashdish-custom-9 | Tell me which of Winstop, Man vs. Fries, McDonald's, Popeyes, and Ike's has the cheapest fries item and its price. | 5 | 8 | Parallel |
| | Gomail | v1/gomail-1 | How many unread emails are in the Inbox? | 1 | 1 | Serial |
| | | gomail-custom-2 | Forward the first two emails in my inbox to me@example.com, each with a short note "Forwarded from my inbox", and then delete the two original emails. | 2 | 12 | Hedge |
| | | v1/gomail-3 | Compose a new email to jonathan.smith@example.com with the subject "Meeting Notes" and body "Please find the meeting notes attached." | 1 | 5 | Serial |
| | | v2/gomail-14 | Clear all my emails that are visible please! | 50 | 3 | Hedge |
| | | gomail-custom-5 | Write an email to each of alice, bob, charlie, david@example.com with personalized greetings and a simple Merry Christmas message. | 4 | 20 | Parallel |
| | | gomail-custom-6 | Reply to each of the first four emails in my inbox with "unsubscribe". | 4 | 20 | Parallel |
| | | gomail-custom-7 | Delete all emails from "Alerts", "Newsletter", or "Notifications". | 13 | 14 | Hedge |
| | | gomail-custom-8 | Create three new labels: "Work", "Personal", "Urgent". | 3 | 9 | Parallel |
| | | gomail-custom-9 | Draft a new email and send it to any four different people. | 1 | 5 | Serial |
| | Omnizon | omnizon-custom-1 | Find the most expensive order on Returns & Orders page and cancel it. | 1 | 5 | Serial |
| | | omnizon-custom-2 | Go to Returns & Orders page and write a product review for each item in the first three orders. | 5 | 26 | Parallel |
| | | omnizon-custom-3 | Cancel the three orders placed in July on the Returns & Orders page. | 3 | 10 | Serial |
| | | omnizon-custom-4 | How many orders have not yet shipped? How many cancelled? | 2 | 3 | Serial |
| | | v1/omnizon-10 | Click on "buy now" on any product, increase its quantity to the maximum allowed, update the delivery date to the last available, and place the order. | 1 | 6 | Hedge |
| | | omnizon-custom-6 | Which of the following coffee makers has touchscreen: Gevi 10, Gevi 12, Simply Good Coffee, and BELLA Single Serve? Visit the product pages of each. | 4 | 12 | Parallel |
| | | omnizon-custom-7 | Visit "Gift", "Toy", "Gaming", and "Cosmetic" categories on the menu bar. For each, click on the first item and report the product color. | 4 | 8 | Parallel |
| | | v1/omnizon-8 | Search for "Automatic Espresso Machine", click on the cheapest one, change the quantity to 5, use "buy now" to purchase them and complete the checkout. | 1 | 7 | Hedge |
| | | v2/omnizon-9 | I want to create a gaming collection—buy any gaming device under $100. | 1 | 6 | Hedge |
| WebArena | GitLab | webarena-747 | Start a private project awesome_web_agents with blank template and add Abishek, Vinta as members. | 3 | 19 | — |
| | | webarena-308 | Tell me who has made the most contributions, in terms of number of commits, to the primer/design project. | 1 | 3 | — |
| | | webarena-414 | Make the LICENSE of byteblaze/dotfiles to MIT license. | 1 | 5 | — |
| | | webarena-133 | How many commits did Eric make to a11yproject on 3/2? | 1 | 3 | — |
| | | webarena-419 | Set my GitLab status as "Enjoying life". | 1 | 4 | — |
| | Reddit | webarena-29 | Tell me the count of comments that have received more downvotes than upvotes for the user who made the latest post on the DIY forum. | 7 | 6 | — |
| | | webarena-727 | Dislike all submissions created by PatientBuilder499 in subreddit videos. | 1 | 7 | — |
| | | webarena-580 | Create a new forum named sci_fi, with a description and include specific sidebar categories. | 1 | 7 | — |
| | | webarena-399 | Change my Reddit bio to "I am a robot". | 1 | 5 | — |
| | | webarena-404 | Upvote the newest post in books subreddit. | 1 | 5 | — |

