# OpenReview forum: "Agent JIT Compilation for Latency-Optimizing Web Agent Planning and Scheduling"
_ICML.cc/2026/Conference — ICML 2026 regular_

### Official Review · Reviewer_BM28 · 2026-02-23

**Soundness:** 2
**Presentation:** 1
**Significance:** 3
**Originality:** 3
**Overall Recommendation:** 3
**Confidence:** 3

**Summary:**

The paper introduces a new system for executing agentic tasks in a browser environment. The core idea is to let an LLM propose a full execution plan to accomplish a task, allowing calls to functions and to LLMs. Functions are specified via an augmented form of MCP, additionally requiring to specify a function's pre- and post-conditions. The proposed plan(s) are subsequently analyzed with respect to correctness using the specified pre- and post-conditions, and an estimation of the plan's execution cost is computed. The cheapest correct plan among a set of proposals is selected and executed. A parallel pipeline determines the execution mode (sequential, parallel, hedged) based on the task description, and a learned cost function. An evaluation with state-of-the-art LLMs shows promising results on the REAL and WebArena benchmark sets.

**Compliance With Llm Reviewing Policy:**

Affirmed.

**Final Justification:**

The presented paper has undoubtably a number of clarity issues, with implications on the reader's ability to confidently justify soundness. However, the authors' response clarified most of the points I was raising, which made me more positive about this paper. I am torn between a rating of 3 and 4, but decided to go with 3 since the changes necessary to incorporate the rebuttal feedback are substantial. The revision should hence go through another round of evaluation.

**Key Questions For Authors:**

1. What is the structure of the execution plan, i.e., what statements are allowed, and how is it executed?
2. How is the cache constructed exactly, what does it contain, how is it invalidated, and how is it used during execution?

**Limitations:**

Yes

**Strengths And Weaknesses:**

The paper proposes the interesting (albeit not very new) idea of improving latency and runtime time of LLM-driven agent for browser tasks into some form of preprocessing. The architecture is novel and the empirical results show considerable advantages against other agentic systems. The topic is clearly relevant for ICML and related work is sufficiently covered.

However, the write-up has severe clarity issues, important details are omitted, and experiments are irreproducible; leaving the paper in a form that is not quite ready yet for publication. The points are spelled out in the following.

A definition of execution plans is not provided, leaving it unclear what kinds of statements are allowed. For example, the use of CFG (control follow) suggests that the plan might contain conditional branching, but it is not clear whether this is actually the case; and if it is, then a sufficiently detailed explanation of the execution engine is missing (in particular, how would the conditions be evaluated). Moreover, a clear definition of what form the pre- and post-conditions need to take is also missing. And with that it is also not clear how validity can be checked; and how the "condition state" update process is supposed to be working. For example, with the missing definition of pre-condition, it is not clear what the check in line 11 of Alg 1 does exactly. The update in line 13 seems to suggest that the post-condition can actually not invalidate anything what was true beforehand, which would significantly limit the expressibility of the annotations.

Besides this core algorithm, the paper further introduces an execution cache and a model evaluating the suitability of the different execution modes for the given task. For both components, the paper lacks sufficient details. For the cache, it is not even clear how it is obtained, how invalidation is handled, whether this is local per task or something learned at global level and used as a database and so on. Regarding the execution mode cost model, the paper lacks any description of how it has been learned. Both are essential parts of the overall architecture, and hence need to be explained in a suitable details level.

The general architecture also seems to be flawed. The best execution mode obviously depends on the specific execution plan; yet the plan is ignored completely when determining the mode. Secondly, it is not clear in how far the execution mode actually guarantee correctness and safety of the execution. For example, the execution plan might contain write operations, inherently necessitating sequential execution. The execution plan might contain side effects (e.g., manipulations of some external system like a shared calendar). This means that statements of the plan might not be executable in parallel; or that even entire plan can be run multiple times in parallel (hedge mode). It is not clear how the proposed system deals with such situations.

The architecture comes with several hyperparameters, yet the authors mention at no point how they are chosen. This pertains to the execution mode cost model, as mentioned above, but also to the cost computation of the execution plans. Furthermore, this computation actually seems to too simplistic. Different options and different prompts certainly come at different costs. So, it is not clear why a constant cost should be used throughout all the different calls.

---

> ### Author Rebuttal · Authors · 2026-03-31
>
> Thank you for the detailed feedback. We recognize that algorithmic details were insufficiently explained and expand below:
>
> **Structure of execution plans**
>
> > "... it is not clear whether \[conditional branching\] is actually the case."
>
> A plan is a block of code that may contain tool calls, `for` loops, `if/else` conditionals, and `ai_eval` calls for inference. The documented CFG traversal walks both branches of conditionals to validate pre/postcondition chains and to estimate cost of a given plan. Please see formal grammar and details in `examples/plan_grammar.md`.
>
> **Pre and postconditions**
>
> > "A clear definition of \[pre/post-conditions\] is missing…"
>
> Preconditions and postconditions are key-value dictionaries over a finite set of state variables (e.g., `page_type`, `selected_item`, `cart_count`) for each tool:
> \- pre: required state before tool execution (e.g., `{page_type: "home"}`)
> \- post: state guaranteed after (e.g., `{page_type: "orders"}`)
>
> Line 11 (validity check): checks that current state satisfies the tool's precondition (key-value subset check). Line 13 (state update): merges the tool's postcondition into the current state via dictionary update. The reviewer is correct that this is a monotonic union. Postconditions overwrite keys they specify but do not remove prior state. This intentionally models a common pattern where a new environment state may e.g. change a `page_type` variable but preserve a `cart_count`.
>
> **Cache construction and invalidation**
>
> Caches are global per environment, not per task. The planner cache maps task patterns to validated code plans and the scheduler cache maps DOM schema elements to latency distributions. Both caches are populated via offline trace collection for tool synthesis with specific tasks that are optional or if set completely distinct from evaluation tasks. Please see our response to Reviewer oJvS ("Tool synthesis") for the trace collection process.
>
> If a tool's precondition or postcondition fails at runtime, the tool’s entries in the planner and scheduler caches are invalidated and the system falls back to planning without it. Alternative invalidation strategies (TTL-based, periodic re-validation) are discussed in `limitations.md`.
>
> **Execution mode cost model**
>
> > "... description of how it has been learned."
>
> The scheduler's cost model uses Monte Carlo estimation over learned latency distributions (Section 3.3). For each DOM schema element, we fit latency distributions from the 10 trace executions per task (see our response to Reviewer oJvS, trace origin section). Given a task, the model predicts which elements the plan will interact with if executed (PREDICTUSAGE, Algorithm 2 Line 10), then estimates total latency for each strategy (serial, parallel, hedge) by sampling from the fitted distributions. The strategy with lowest estimated latency is selected.
>
> **Hyperparameters**
>
> > "... how they are chosen."
>
> In the planner (Algorithm 1): the key hyperparameter design choice is $C_\text{tool}=0.1 \ll C_\text{eval}=10.0$. Plans using cached tools are strongly preferred over those invoking LLM calls. Since the cost model ranks candidates rather than predicting absolute latency, the exact values are less significant. This is supported by the results showing the true latency difference between candidates with best and worst estimated cost..
>
> In the scheduler (Algorithm 2): hyperparameters (strategy overheads, page read costs) may be profiled per model on the same tasks for collecting traces for offline tool synthesis.
>
> All hyperparameter values were set based on initial experiments on the trace collection tasks and kept fixed across all evaluation tasks, models, and environments. We did not tune or perform a hyperparameter sweep that could potentially confound results. Please see `appendix/hyperparameters_and_cost_model.md`.
>
> **Architecture: planner and scheduler separation**
>
> > "The best execution mode depends on the plan; yet the plan is ignored when determining the mode..."
>
> The JIT-Planner and JIT-Scheduler are separate components that address distinct aspects of the latency problem and thus are evaluated separately. The planner orchestrates tool usage and LLM inference while the scheduler allocates vCPUs. They are introduced as complementary methods for agent JIT optimization.
>
> The reviewer is correct that the scheduler could benefit from seeing the plan. Currently the scheduler predicts element usage from the task alone, which is available immediately without waiting for planning, but a plan-aware scheduler is a natural future direction.
>
> Regarding parallel/hedge execution with writes: safe parallelism is not a core contribution, but it is practical for most applications. See our response to Reviewer Ua92 ("Parallel candidate generation").
>
> Our supplementary repository is available at [https://anonymous.4open.science/r/agent-jit-icml-supplement-789D/](https://anonymous.4open.science/r/agent-jit-icml-supplement-789D/).

---

> > ### Author Rebuttal · Reviewer_BM28 · 2026-04-02
> >
> > Thanks for the answer. I tend to leave my rating unchanged given the deficiencies in the current write-up (which actually have also been pointed out by the other reviewers). This is definitely an interesting piece of work, but the descriptions need to be polished considerably. Moreover, the exact reasoning procedure dealing with the pre- and post-conditions should be explained more thoroughly. Program analysis is in general a highly complex (depending on the properties and allowed statements, potentially even undecidable) computation problem. It is hence important to lay down precisely the form of programs dealt with in this paper and the conditions and properties that are considered, in order to precisely understand how this challenging problem tackled.

---

> > > ### Author Response · Authors · 2026-04-08
> > >
> > > Thank you for the acknowledgement having concerns adequately addressed.
> > >
> > > We agree that program analysis is generally complex and sometimes undecidable. Our system mitigates this by restricting both what programs can look like (program structure) and what properties are checked (property language):
> > >
> > > **Restricted program structure.** Our plans are a strict subset of Python: sequential tool calls, for/while loops, if/else conditionals, and `ai_eval` calls. Tool names are resolved statically from the AST (no dynamic dispatch, no higher-order functions). For loop constructs, the validator unrolls the body up to 2 iterations to verify that tool preconditions hold across iterations. This bounds the CFG to a finite size. In practice, this suffices because tool pre/post conditions are static at runtime: if the state after one iteration satisfies the preconditions for re-entry, all subsequent iterations will too. The validator handles conditional branches within loops correctly by forking state at each if/else and checking re-entry from all possible end states (see our detailed example in `appendix/formal_analysis.md`). The bound is also reconfigurable.
> > >
> > > **Restricted property language.** Pre/postconditions are key-value dictionaries over a small set of state variables (typically 2–5 per application, e.g., `page_type`, `selected_item`). Checking involves verifying specific keys have specific values. There are no arithmetic constraints, quantifiers, or temporal properties to reason about. The state update is a dictionary merge.
> > >
> > > **Validation algorithm.** The algorithm makes a single forward pass over the CFG. At each tool call: (1) check that each key in `tool.pre` matches the current state, (2) merge `tool.post` into state. For if/else, fork the state and validate both branches. This doesn’t involve backtracking or constraint solving, and the algorithm is linear in the number of CFG blocks. In our benchmarks, validation takes 0.06ms for a 5-tool plan and 0.12ms for a 9-tool plan with nested loops.
> > >
> > > Our implementation of the analyzer consists of three modules totaling ~ 1,750 lines of Python: CFG construction from the Python AST (~ 450 LoC), pre/postcondition validation via CFG traversal (~ 880 LoC), and cost estimation via CFG traversal (~ 410 LoC). We have found our implementation to work reliably across all tasks. We release the source code and hope the community will find it useful.
> > >
> > > To precisely lay down the form of programs and the conditions/properties considered, we have added the following to our supplementary repository (https://anonymous.4open.science/r/agent-jit-icml-supplement-789D/):
> > > Detailed description of the formal grammar and validation algorithm (under `appendix/formal_analysis.md`), and
> > > Analyzer source code (under `src/`).
> > >
> > > These complement existing code plan examples (`appendix/planner_examples.md`) and the full synthesized tool registries for all 5 applications with their actual pre/postconditions (`tools/`).
> > >
> > > We appreciate the reviewer's feedback and will incorporate precise specifications as well as examples into the appendix of the revised paper.

---

### Official Review · Reviewer_Ua92 · 2026-03-12

**Soundness:** 4
**Presentation:** 3
**Significance:** 4
**Originality:** 4
**Overall Recommendation:** 5
**Confidence:** 4

**Summary:**

The paper proposes an algorithm to speed up browser-operating LLM agents by collecting, maintaining and applying a cache of programmatic actions to replace lengthy LLM steps. Experiments on 5 environments vs 4 baselines show an order of magnitude acceleration along with the score improvement.

**Compliance With Llm Reviewing Policy:**

Affirmed.

**Key Questions For Authors:**

[1] “Parallel candidate generation” section. How is parallel (vCPU) schedule exploration possible for a single browser session? Is the session replicated (cloned) or rolled back?

[2] Please provide several examples of the learnt JIT programs.

**Limitations:**

[1] Technical complexity of the implementation may hinder practical applications.

[2] It is not clear how the algorithm can be run on stateful environments that do not allow roll-backs or cloning.

**Strengths And Weaknesses:**

# Strengths

[1] The idea to shortcut expensive LLM calls with a set of dynamically collected micro-programs is novel and bright. Introducing this type of test-time/online learning is meaningful and consequential.

[2] Maintaining a database (cache) of scheduling primitives and their costs (durations) is a good idea.

[3] The end-to-end latency numbers (Figure 6) are extraordinary.

[4] The environment vs baseline matrix is big enough and convincing.

[5] The proposed algorithm is to some extent neuromorphic: living organisms solve complex tasks with their prefrontal cortex, but once tasks become known and routine, they are pushed to the cerebellum and the spinal cord.


# Weaknesses

[1] The setup and the algorithm are quite complex.

[2] The term “Scheduler Cache” is somewhat misleading, since it is rather an experience buffer, or a database.

---

> ### Author Rebuttal · Authors · 2026-03-31
>
> We thank the reviewer for the positive assessment and the insightful neuromorphic analogy. We address the two questions and limitations below:
>
> **Parallel candidate generation for a single browser session**
>
> > "How is parallel (vCPU) schedule exploration possible for a single browser session?"
> > "... how the algorithm can be run on stateful environments that do not allow roll-backs or cloning."
>
> There are two instances of parallelism in our system:
>
> * Parallel candidate generation (Algorithm 1) operates at the *planning* stage. Multiple workers independently sample code plans from the LLM in parallel.
> * Parallel/hedge execution (Algorithm 2) does require environment clones. We launch each parallel environment clone from the same initial state (same URL, same cookies via Playwright's `storage_state`), and each executes independently. We do not assume the ability to roll back. For environments where parallel environments are not feasible, the system falls back to serial execution.
>
> Regarding stateful environments with write operations: many web applications follow a draft-then-commit pattern. E-commerce sites maintain per-session carts (isolated by default), email clients compose drafts without sending, and food delivery apps hold orders in a pending state until explicit confirmation. In hedge mode, each worker executes up to the commit point (e.g., fill cart, compose email), and only the fastest worker commits; the rest are discarded (optimistic concurrency control). In parallel mode, each worker handles an independent subtask (e.g., "reply to email 1", "reply to email 2"), and all results are collected and verified before a group commit.
>
> For environments more generally such as desktop applications, cloning is possible via VM snapshotting (e.g., QEMU).
>
> We note that the proposed planner and scheduler are separate components that can be adopted independently. See our response to Reviewer BM28 ("Architecture") for further discussion.
>
> **Examples of learnt JIT programs**
>
> > "Please provide several examples of the learnt JIT programs."
>
> The following are three code plans from actual runs on: *"Visit the first five restaurants in 'Best near you' and report how many have under 20 reviews."*
>
> * Plan A — Rejected. Calls `get_store_details()` directly after `list_all_stores()`, but `get_store_details` requires `pre.page_type = "store"` while the state is still `"home"`. The protocol rejects this plan at validation time (Algorithm 1, line 12).
> * Plan B — Valid but suboptimal. Correctly computes the count in pure code, but then wraps it in an unnecessary `ai_eval(...)` call. Cost: $C_\text{tool} + C_\text{eval} = 0.1 + 10.0 = 10.10$.
> * Plan C — Optimal (selected). Same logic as B but formats the result as an f-string, eliminating the LLM call entirely. Cost: $2 × C_\text{tool} = 0.20$. The planner selects Plan C (\~50× cheaper than B).
>
> Here are two examples for scheduler strategy selection:
>
> * Example 1 — Serial selected. Task: *"How much is a matcha croissant and a regular iced matcha latte from Stonemill Matcha?"* Only one click needed (`restaurantCard`, mean 9.3s, σ=2.4s), then all prices are readable from the page. MC result: Serial 19.1s (72.2% win rate) vs Hedge 21.2s vs Parallel 63.0s. Scheduler selects *Serial*.
> * Example 2 — Hedge selected. Task: "*Place an order for any type of sub-sandwich, keep the total under $30."* This needs six sequential interactions across 4 pages and is not parallelizable. The bottleneck is `fullMenuItemAddButton` (mean 24.8s, σ=25.8s) — locating the right button on a crowded menu page is highly unpredictable, with observed latencies from 6s to 91s. Hedging runs 4 replicas and takes the fastest, which helps precisely because of this variance. MC result: Hedge 114.7s (72.6% win rate) vs Serial 133.5s. Scheduler selects *Hedge*.
>
> **Technical complexity**
>
> > "Technical complexity … may hinder practical applications."
>
> The components we propose each address a distinct concern (protocol for plan correctness, planner for cost-optimal tool orchestration, and scheduler for vCPU allocation) and can each be adopted independently. For example, the invariant-enforcing protocol alone improves accuracy by 10–17 percentage points (Table 1\) and can be added to any existing tool-use agent without the planner or scheduler.
>
> **Naming: "Scheduler Cache"**
>
> > "The term 'Scheduler Cache' is somewhat misleading, since it is rather an experience buffer, or a database."
>
> We appreciate this suggestion. Our naming rationale was that what it stores is possible to invalidate. "Experience buffer" does capture the fact that the scheduler cache stores learned latency distributions from prior execution. We will consider this terminology in the revision.
>
> Our supplementary repo is available at [https://anonymous.4open.science/r/agent-jit-icml-supplement-789D/](https://anonymous.4open.science/r/agent-jit-icml-supplement-789D/).

---

> > ### Author Rebuttal · Reviewer_Ua92 · 2026-04-01
> >
> > Thank you for your answers and especially for the examples. I am keeping my score as Accept.

---

### Official Review · Reviewer_6Js8 · 2026-03-13

**Soundness:** 2
**Presentation:** 3
**Significance:** 4
**Originality:** 3
**Overall Recommendation:** 5
**Confidence:** 3

**Summary:**

This paper introduces an alternative agent loop for computer-use agents, JIA compilation, that enables higher latency and accuracy. This is achieved through an invariant enforcing tool protocol which checks the validity of plans, a JIT-Planner which generates multiple plans and chooses the lowest cost one, and a JIT-scheduler which choose the lowest cost strategy from serial, parallel, and hedge strategies. The JIT-Planner achieves a minimum of 7.4x latency improvements to up to 19.1x with large accuracy improvements (from 13% to up to 38%).

**Compliance With Llm Reviewing Policy:**

Affirmed.

**Final Justification:**

The rebuttal has made it clearer that the contribution of the paper was the latency gains, which were impressively high. The authors have also agreed to revise the title of their work to accurately reflect the current evaluation benchmarks, so I have decided to increase my score to Accept.

**Key Questions For Authors:**

While the results are impressive, there is some missing information in the paper that I need clarified to understand the practicality of applying this for general computer-use agents.

(1) How are the execution traces that are used to populate the planner cache retrieved? This data provides synthesized tools for the Browser-Use + cache baseline and is provided in context to the LLM when generating candidate plans and choosing a scheduling strategy.

(2) What do the prompts look like for the LLM? Ideally they should all be included in the Appendix but I am interested in seeing the (1) parallel candidate generation and the (2) PREDICTUSAGE prompt in Algorithm 2 Line 2 to understand how the LLMs make use of the cache data.

(3) How were the invariant-enforcing tool protocols created for each tool? Given general computer-use agents would be working in stochastic environments, how is this considered for post-condition generation?

(4) The abstract and title seem to imply this paper focuses on general computer-use agents but the remainder of the paper reads like a web agent paper (e.g. invariant-enforcing tool protocol is for a web page, every evaluated application is for web-pages). Does this method additionally apply for general computer-use and if so how would the invariant-enforcing tool protocol be leveraged in the absence of a11y tree data (e.g. tools for image processing on GIMP, debugging a program on VS Code, or even to play an executable game)?

**Limitations:**

The limitations do not seem to be discussed; one clear limitation of this work is that the invariant-enforcing tool protocol can be brittle in stochastic environments since post-conditions may not be guaranteed.

**Strengths And Weaknesses:**

The empirical results are very strong in terms of high accuracy and lower latency gains. The method seems mostly reasonable but in practice the invariant-enforcing tool protocol would likely be ineffective for some computer-use tasks since it requires pre conditions which may not be reachable (if no tool can take you from the current initial state to a known pre-condition state) and a post-condition that certainly cannot always be guaranteed in stochastic environments (e.g. any kind of pop-up ad or bot detector). In addition, the process for obtaining the execution traces for populating the planner cache is absent from the paper. The figures and differentiation from past works in the paper is very clear. However, there is some missing information as mentioned above. This paper addresses the core agent loop in computer-use agents with a new general method to improve latency which could be applied to all computer-use agent works.

---

> ### Author Rebuttal · Authors · 2026-03-31
>
> **Execution traces for planner cache**
>
> > "How are the execution traces that are used to populate the planner cache retrieved?"
>
> Please see our detailed response in the "Tool synthesis and evaluation fairness" section of our response to Reviewer oJvS. In short, the method of collecting execution traces for offline tool synthesis is orthogonal to our core contributions and in our system it is an implementation of LATM (Cai et al., 2024\) and WALT (Prabhu et al., 2025). To isolate the impact of our core contributions, across 5 environments, we use either 2 or 3 held-out tasks per environment or no specific tasks at all for collecting traces for offline tool synthesis to populate the planner cache and critically, we ensure that held-out tasks for tool synthesis are distinct from tasks for evaluation.
>
> **LLM prompts**
>
> > "What do the prompts look like for the LLM? Ideally they should all be included in the Appendix but I am interested in seeing the (1) parallel candidate generation and the (2) PREDICTUSAGE prompt in Algorithm 2 Line 2."
>
> We have included all full prompt templates (parallel candidate generation, PREDICTUSAGE, and tool synthesis) in the supplementary repository. Please see the `prompts/` directory.
>
> **Invariant-enforcing tool protocol creation**
>
> > "How were the invariant-enforcing tool protocols created for each tool? Given general computer-use agents would be working in stochastic environments, how is this considered for post-condition generation?"
>
> The pre/postconditions required by the tool protocol are generated alongside tools during offline synthesis. Offline tool synthesis is extensively studied in prior art by LATM (Cai et al., 2024\) and WALT (Prabhu et al., 2025), and the addition of the protocol-required pre/postconditions is less complex (even if quantified by LOC) to reliably generate compared to the tool itself. The contribution in our work is designing the protocol to achieve Pareto frontier latency-accuracy improvement for agents.
>
> Regarding stochastic environments: postconditions check *observable outcomes* (e.g., "did the page navigate to the orders page?" or "does the cart count increase?") rather than exact UI state. This makes them robust to incidental variation (e.g., different ad placements, layout shifts). When a postcondition fails due to genuine stochasticity (e.g., a bot detector), the system treats this as a tool failure and falls back to tool invalidation and generic primitive tools: see our response to Reviewer BM28, "Cache construction and invalidation".
>
> We acknowledge this as a limitation for highly stochastic environments and discuss it in the revised Limitations section.
>
> **Applicability beyond web**
>
> > "The abstract and title seem to imply this paper focuses on general computer-use agents but the remainder of the paper reads like a web agent paper."
> > "Does this method additionally apply for general computer-use and if so how would the invariant-enforcing tool protocol be leveraged in the absence of a11y tree data?"
>
> Our current system implementation and evaluation focus on web applications as the environments the agent operates in, where the DOM provides structured state for pre/postcondition checking. Our proposed approach is not limited to web applications. The key requirement is \*\*programmatic access to environment state\*\* for pre/postcondition evaluation.
>
> For desktop applications, this is possible via robust and well-documented accessibility APIs (available on macOS, Windows, Linux), application-specific APIs and file system state. We have not yet evaluated desktop applications or similar environments and acknowledge this as future work.
>
> **Limitations**
>
> > "The limitations do not seem to be discussed; one clear limitation of this work is that the invariant-enforcing tool protocol can be brittle in stochastic environments."
>
> We have added a dedicated Limitations section to the revised paper covering: (1) offline setup cost for tool synthesis, (2) cache staleness under UI changes, (3) brittleness of the invariant-enforcing protocol in stochastic environments, and (4) evaluation scope (37 tasks, web-only). A detailed write-up is available at [https://anonymous.4open.science/r/agent-jit-icml-supplement-789D/limitations.md](https://anonymous.4open.science/r/agent-jit-icml-supplement-789D/limitations.md).
>
> Our supplementary repo is available at [https://anonymous.4open.science/r/agent-jit-icml-supplement-789D/](https://anonymous.4open.science/r/agent-jit-icml-supplement-789D/).

---

> > ### Author Rebuttal · Reviewer_6Js8 · 2026-04-04
> >
> > Thank you for the response. Given the caching mechanism is prior work and the contribution is the low latency (with lack of degradation compared to the ablation Browser-use + cache), I am willing to raise my score to an Accept. However,
> > > We have not yet evaluated desktop applications or similar environments and acknowledge this as future work.
> >
> > would it not be appropriate to rename the paper to "Agent JIT Compilation for Latency-Optimizing Web Agent Planning and Scheduling"? If this is to be delegated to future work, the current title as it stands is misleading. I agree with the authors, it would be trivial to apply this framework to computer-use benchmarks; however, there are no results on them.

---

> > > ### Author Response · Authors · 2026-04-04
> > >
> > > Thank you for the follow-up and for being willing to raise your score.
> > > > would it not be appropriate to rename the paper to "Agent JIT Compilation for Latency-Optimizing Web Agent Planning and Scheduling"?
> > >
> > > We agree the title should reflect the current scope of evaluation and we will rename the paper to "Agent JIT Compilation for Latency-Optimizing Web Agent Planning and Scheduling" in the revision.

---

### Official Review · Reviewer_oJvS · 2026-03-16

**Soundness:** 2
**Presentation:** 3
**Significance:** 2
**Originality:** 2
**Overall Recommendation:** 4
**Confidence:** 3

**Summary:**

This paper identifies a current research and engineering gap: many publicly available computer use agents (CUAs) have a fairly simple architecture, and exhibit poor latency and task performance. The paper presents latency optimising approaches for CUAs, focusing on the planning and task scheduling stages. They improve the latency of current CUAs by introducing a cost-optimising planner and a cost-aware scheduling module into their agent system, (essentially a module which translates tasks in natural language into executable code plans, which are then verified, cached, and compared to find the one with optimal latency) and maintain or improve accuracy by introducing invariant pre and postcondition state validation to avoid generating or executing plans with incorrect steps. They evaluate their system on 37 tasks across 5 applications. Their JIT-planner reportedly achieves up to a ~7x speedup compared to the next best baseline (Browser-Use+cache), as well as a +28% accuracy improvement.

**Compliance With Llm Reviewing Policy:**

Affirmed.

**Final Justification:**

The rebuttal resolved my two most serious concerns  and the new CUA+Tools ablation is a strong addition. Raising from 3 to 4.

**Key Questions For Authors:**

- Where do the execution traces for offline tool synthesis come from? Are they collected by running the agent on the same test tasks, or a separate set of tasks within the same applications? What model are the traces generated by? How many are collected? If traces are from the test tasks, can the authors report accuracy on held-out tasks within the same applications to demonstrate that the approach generalises beyond the tasks used for tool synthesis?
- Can you provide confidence intervals on the scores and speedups?

**Limitations:**

The authors do not discuss limitations in nearly enough detail in the paper. There is also no section dedicated to the discussion of limitations. There are several key weaknesses listed above that require either clarification, or the acknowledgement of the limitation.

**Strengths And Weaknesses:**

Strengths:
- **The authors identify a legitimate problem and present it clearly, effectively motivating their approach.**
- **The method achieves a non-trivial speedup compared to the baseline whilst maintaining or even improving accuracy.**
- **The research questions and ablations are clearly presented, and thorough.**
- **Good choices of metric:** In particular, the pass@k and pass@t metrics for planning efficiency are clearly defined and cleverly operationalised.
- **The paper is clearly presented**, with architecture graphs, clear input/output examples, readable pseudocode for the two algorithms, and visualisations of key results.
- **The paper explores a range of scheduling strategies, and identifies domains in which each of them excel.** The analysis is clear, and the identification of success/failure patterns could help further optimise CUAs in the future if a dynamic dispatcher is used.
- **Task cardinality is a principled and objective approximation of task difficulty:** It is good that the authors avoid falling back on subjective metrics here.
- **The oracle scheduler is a strong and well formulated upper bound.** It gives a clear idea of how much performance optimisation is left on the table, and how much the paper's JIT system closes the gap.

Weaknesses:
- **Very small task set size:** the whole evaluation suite only contains 37 tasks across 5 apps, making the results limited for drawing generalisable conclusions. The authors could improve this by measuring on a wider variety of tasks, and by clarifying if they ran repeat samples on any of the tasks to ensure statistical significance.
- **JIT language is misleading:** the JIT compilation language, borrowed from CS literature, is a weak analogy to an actual jit compiler, and the "compilataion" modules are largely controlled by prompted LLM modules writing (afaict) python code, and latency optimisation modules.
- **The comparison to the baselines is partially unfair:** As far as I understand, the JIT system uses pre-built, cached, and application specific tools, while the non-cached baselines use generic primitives to take actions. The authors do include a Browser-Use +cache ablation which gives the baseline access to the same synthesized tools, and the JIT-Planner still achieves ~7x speedup over this, which suggests the compilation/planning layer adds genuine value beyond just having better tools. However, the CUA baselines (Anthropic, OpenAI) are not given access to the same tools, making those comparisons less clean. I also expect that, if you used a wider variety of tasks that were further out of distribution from the computer use training, tools using more generic primitives might excel in accuracy.
- **The source of the execution traces used to generate pre-cached tools is unclear:** This is a significant limitation. It's unclear how similar the tasks or environments used to pre-generate agent tools are to the environments being tested. If the traces are collected on the test tasks themselves, then this is a massively unfair advantage for this system over others. The tools are essentially guaranteed to cover exactly the actions needed to complete the evaluation, which would not be the case in novel, real world computer use applications.
- **No statistical significance tests or confidence intervals reported.**
- **Unclear if pre-built or cached tools are robust to App/UI changes** - is this approach possible in a real deployment, where UIs might be changing rapidly? Or will performance of the system degrade over time?
- **Offline setup costs are unreported:** There appear to be some steps (gathering traces, pre-building tools) that are run offline, within each application environment, before evaluation time. What is the cost of these steps relative to the evaluation itself? How scalable is the method to other, more complex, application environments?

---

> ### Author Rebuttal · Authors · 2026-03-31
>
> We thank the reviewer for the constructive feedback. We address the concerns below:
>
> **Tool synthesis and evaluation fairness**
>
> Since offline tool synthesis is prior art and not part of our contributions, we (1) applied prior art for offline tool synthesis and (2) designed the evaluation to prevent the efficacy of tool synthesis from confounding results. For 3 of the environments, we apply LATM (Cai et al., 2024\) where tools are synthesized from traces generated by executing tasks that are completely distinct from evaluation tasks. For 2 of the environments, we apply WALT (Prabhu et al., 2025\) where traces are generated by autonomous exploration of the environment. This careful evaluation setup (1) demonstrates that specific tasks for tool synthesis are optional and (2) if specified they are completely distinct from evaluation tasks.
>
> Hence, we isolate the impact of our core contributions which are orthogonal to offline tool synthesis: the invariant-enforcing tool protocol and cost-optimizing planning and scheduling. For example, the Browser-Use \+cache baseline has the same tools but lacks planning and we find it to be \~7× slower.
>
> **Baseline fairness**
>
> > "CUA baselines … are not given access to the same tools"
>
> We agree and have added a CUA \+ Tools ablation using GPT-5.4 (OpenAI's most capable CUA model) with access to the same synthesized tools. This isolates the benefit of cost-aware planning from access to synthesized tools.
>
> JIT-Planner provides a consistent 1.5-2.4× additional latency reduction at comparable accuracy (\~95%) across all cardinality and length tiers, despite using a much less capable model (GPT-4.1) not fine-tuned for computer use. Full table and breakdowns by cardinality/length are in `appendix/cua_ablation.md`.
>
> > "I also expect that, if you used a wider variety of tasks that were further out of distribution from the computer use training, tools using more generic primitives might excel in accuracy."
>
> We agree generic primitive tools could generalize more than synthesized tools. That said, we note a few things that mitigate this in practice:
>
> First, prior art for offline tool synthesis such as LATM (Cai et al., 2024\) and WALT (Prabhu et al., 2025\) synthesize task-general tools (navigation, listing, forms, search) rather than task-specific ones, even when generated from tasks distinct from those for evaluation. For example, Browser-Use \+cache, which uses the same synthesized tools, achieves 80–100% accuracy across environments.
>
> Second, real-world deployments often adapt to a specific environment (e.g., back-office robotic process automation of a company’s specific internal software), motivating prior art for offline tool synthesis. Our contributions achieve a Pareto frontier improvement orthogonal to the method for offline tool synthesis and fairly compare.
>
> **Statistical rigor**
>
> We report 95% CIs and significance tests for all key claims. Full tables are in `statistical_rigor.md`.
>
> All major claims are significant: JIT-Planner achieves 10.4× \[9.0×, 12.8×\] speedup and \+28 pp \[+19, \+34\] accuracy vs Browser-Use (both p \< 1e-7). Best-vs-worst cost selection confirms the cost model works: 1.8× \[1.6×, 2.1×\] (p \= 4.6e-12). JIT-Scheduler provides 1.8–2.4× speedup vs OpenAI CUA (p \< 0.003) and \+16–20 pp accuracy vs Serial for GPT-4.1 and Gemini-2.5-Pro (p \< 0.02). Protocol enforcement adds \+10.7–16.8 pp accuracy across all models (all p \< 1e-10).
>
> **Task set size**
>
> Our 37 tasks span 5 representative environments across 2 established benchmarks and vary tasks across multiple axes (cardinality, length, environment type). As an academic lab constrained by inference costs, we designed the evaluation for high coverage per dollar. We ran 3 repeat trials per task configuration, and all major claims are statistically significant.
>
> **Cache invalidation**
>
> Please see BM28 ("Cache construction and invalidation") regarding robustness to UI changes.
>
> **Offline setup costs**
>
> In our system implementation where we apply the approaches of LATM (Cai et al., 2024\) and WALT (Prabhu et al., 2025\), tool synthesis takes 25–90 min per environment and trace collection is 25–45 min, but both are parallelizable across workers to \~20–30 min per environment. These are one-time costs amortized over subsequent tasks. See `limitations.md` for per-environment breakdown.
>
> **JIT naming**
>
> Like a JIT compiler, our approach just-in-time (at run time) selects the cost-optimal translation of higher-level instructions (natural language) into lower-level code (orchestrating tool calls and inference). We acknowledge the analogy is imperfect and are happy to soften the language in the revision.
>
> **Limitations section**
>
> Please see `limitations.md`. Our supplementary repository is available at [https://anonymous.4open.science/r/agent-jit-icml-supplement-789D/](https://anonymous.4open.science/r/agent-jit-icml-supplement-789D/).

---

> > ### Author Rebuttal · Reviewer_oJvS · 2026-04-06
> >
> > Thank you for your response. Given that you have convincingly addressed most of my claims, I'm willing to bump my score up to a weak accept.

---

### Decision · Program_Chairs · 2026-04-30

**Decision:**

Accept (regular)

**Comment:**

The paper presents a method for planning and scheduling in AI agent situations, including resources and tools. The general idea is to generate code directly for a policy.

The idea is interesting and the scope is probably valuable.

It seems many reviewers had issues with the manuscript and there are still some unsolved questions. At least one review was doubtful that we can imagine how are improved manuscript would look like. I agree with that reviewer. I think the paper deserves another iteration, making things clearer, upfront. Given the support the paper receives, I’m callibrating my recommendation to Weak Accept.